# A data-independent acquisition-based global phosphoproteomics system enables deep profiling

Reta Birhanu Kitata [1], Wai-Kok Choong [2], Chia-Feng Tsai[3], Pei-Yi Lin[1], Bo-Shiun Chen[1,4], Yun-Chien Chang[1,4], Alexey I. Nesvizhskii [5], Ting-Yi Sung[2] & Yu-Ju Chen [1,4 ✉]

Phosphoproteomics can provide insights into cellular signaling dynamics. To achieve deep and robust quantitative phosphoproteomics profiling for minute amounts of sample, we here develop a global phosphoproteomics strategy based on data-independent acquisition (DIA) mass spectrometry and hybrid spectral libraries derived from data-dependent acquisition (DDA) and DIA data. Benchmarking the method using 166 synthetic phosphopeptides shows high sensitivity (<0.1 ng), accurate site localization and reproducible quantification (~5% median coefficient of variation). As a proof-of-concept, we use lung cancer cell lines and patient-derived tissue to construct a hybrid phosphoproteome spectral library covering 159,524 phosphopeptides (88,107 phosphosites). Based on this library, our single-shot streamlined DIA workflow quantifies 36,350 phosphosites (19,755 class 1) in cell line samples within two hours. Application to drug-resistant cells and patient-derived lung cancer tissues delineates site-specific phosphorylation events associated with resistance and tumor progression, showing that our workflow enables the characterization of phosphorylation signaling with deep coverage, high sensitivity and low between-run missing values.

[1] Institute of Chemistry, Academia Sinica, Taipei 11529, Taiwan. [2] Institute of Information Science, Academia Sinica, Taipei 11529, Taiwan. [3] Biological Sciences Division, Pacific Northwest National Laboratory, Richland, Washington 99354, USA. [4] Department of Chemistry, National Taiwan University, Taipei 10617, Taiwan. [5] Department of Computational Medicine and Bioinformatics, and Department of Pathology, University of Michigan Medical School, Ann Arbor, Michigan 48109, USA. ✉email: yujuchen@gate.sinica.edu.tw

Over two-thirds of human proteins are estimated to be phosphorylated resulting in hundreds of thousands of potential sites in a cell to regulate protein function, cellular signaling, and their subversions in human disease[1]. Advances in data-dependent acquisition (DDA) mass spectrometry (MS) have revealed a system view of carrier proteins, sites, and expression levels of site-specific phosphorylation-mediated networks beyond simple abundance-based models[2]. Combining phosphopeptide enrichment, a single-shot DDA-based phosphoproteomics approach achieved different profiling coverage, such as the identification of 12,799 phosphosites from prostaglandin E2-stimulated Jurkat T cells[3] and 13,000 accurate (class 1) sites per 90 min gradient in glioblastoma cells, in a recently reported EasyPhos protocol[4].

To achieve a deep phosphoproteomics depth, extensive peptide fractionation is commonly adapted. A pioneering study by Olsen et al.[5] identified 6600 phosphosites on 2244 proteins, which revealed dynamic epidermal growth factor (EGF) signaling networks. Impressive coverage of >20,000 phosphosites was continuously updated to enhance our understanding of the composition of the phosphoproteome in biological systems[6,7]. Previous deep profiling in HeLa cells achieved 38,229 phosphosites by acquiring ~270 liquid chromatography-tandem MS (LC-MS/MS) datasets measured for about 40 days[8]. Nevertheless, genome-wide phosphoproteome profiling of patient-derived tissues provides insight into how genetic alterations affect the phosphoproteomics landscape in cancer[9,10]. The above phosphoproteomics depth was achieved by the cost of extensive fractionation and days of data acquisition from a large amount of samples, posing limitations to samples with a low cell number or minute clinical specimens.

Semistochastic sampling in DDA also causes quantification challenges, including low phosphoproteome coverage, batch effects, and many missing values across a large number of patients[11]. Data-independent acquisition (DIA), in which all precursors within isolation windows covering the specified $m/z$ range are fragmented, has become an attractive alternative, as DIA enables the retrospective interrogation of preserved fragments to derive peptides from spectral libraries[12]. At present, however, only a limited number of studies have applied DIA for large-scale phosphoproteomics analysis. Parker et al.[13] reported the pioneering DIA-based quantification of 86 phosphoproteins in insulin signaling. Performance of DIA for quantification of targeted phosphopeptide has demonstrated good correlation with the highly specific selected reaction monitoring MS technique[14]. Lawrence et al.[15] also reported large-scale phosphoproteome database as a resource for targeted quantification. Most significantly, DIA demonstrated capability in the differentiation and quantification of positionally isomeric phosphopeptides[16]. Recently, to address the challenge in site-specific analysis of posttranslational modification dataset, various algorithms have been introduced including Inference of Peptidoforms[17], Thesaurus[16], and PIQED[18]. For global phosphoproteomics, Olsen and colleagues[19] recently reported the quantification of >29,000 phosphopeptides (~14,000 localized phosphosites) by a fast LC and DIA method along with phosphosite localization strategy. To approach the disease-associated proteins usually present in low abundance, especially in small-scale clinical specimens, deeper phosphoproteome profiling towards the genome-wide depth still remains to be further developed.

In this study, we report a global phosphoproteomics system (GPS) strategy based on DIA-MS with direct DIA (dirDIA) and library-based (libDIA) computation mapping to high-quality hybrid spectral library derived from DDA and DIA data. By model study on non-small cell lung cancer (NSCLC), analytical merits were benchmarked using 166 synthetic phosphopeptides relevant in lung cancer signaling. Using lung cancer cell lines and patient-derived tissues, we establish a proteome spectral library (12,344 protein groups, 223,091 peptide sequences) and a phosphoproteome spectral library of 159,524 phosphopeptides on 8805 protein groups covering 88,107 phosphosites with increased tyrosine phosphorylation (pTyr; 5483 pTyr, 6%). Overall, the GPS strategy using a single-shot DIA achieves deep quantification of 38,255 phosphosites (20,420 class 1 sites) with 95% unique phosphosites covered by the library-based approach. Application to cell lines and patient tissues further reveals advantages of significantly lower between-run missing values, especially for pTyr, and high sensitivity with deep coverage.

## Results

**Workflow of GPS strategy by DIA-MS.** To achieve deep and highly accurate phosphosite quantification by DIA-MS, a quality phosphopeptide reference library[20] is critical for targeted phosphopeptide signal extraction, while direct data deconvolution of DIA data can be applied[21]. With fast data acquisition by the Orbitrap MS instrument, we show that a comprehensive hybrid phosphoproteomics spectral library resource can serve as a digital map to recover phosphopeptides in the $m/z$ and retention time domains of DIA data. Taking advantage of different data generation for precursors and fragmentation patterns in DIA and DDA, a phosphopeptide library was uniquely constructed in the hybrid mode by complementary datasets from fractionated phosphopeptides. Using NSCLC as a model, several strategies were additionally adapted to enhance identification of low-abundant and cancer-relevant phosphoproteins (Fig. 1a). These included (1) using complementary sample types of NSCLC cell lines and pooled tumor tissues from NSCLC patients with varying EGF (*EGFR*) mutation statuses (Supplementary Data 1); (2) using high pH reversed-phase (HpRP) chromatography for fractionation of tryptic peptides from pooled tissue (column)[10] and individual cell lysate (StageTip)[22], followed by phosphopeptide enrichment using immobilized metal affinity chromatography (IMAC) in the StageTip protocol; and (3) using pervanadate (PV) phosphatase inhibitor treatment to enhance coverage of pTyr. All raw files were acquired by spiking indexed retention time (iRT) peptides and employing consistent chromatography and MS acquisition platforms to maintain uniformity in peptide retention features and fragmentation profiles.

A spectral library was then constructed by Spectronaut (Fig. 1b). To generate the hybrid phosphoproteome library, combined database search results from DDA and DIA dataset were performed by Spectronaut Pulsar, to ensure identification confidence of 1% false discovery rate (FDR) cutoff (peptide spectrum match (PSM), peptide, and protein, and for estimation of phosphosite localization probability[23,24]). For an independent proteome-level library, a similar workflow without Fe-IMAC was performed for NSCLC cell lines and tissue samples. All proteome DDA raw files were searched by MaxQuant with a FDR of 1% at PSM and protein levels (Fig. 1b). For optimal DIA analysis, the data acquisition parameters, including isolation window, resolution, and LC-MS/MS gradient time, were evaluated (Supplementary Fig. 1). The raw DIA data were processed by both libDIA and library-free (dirDIA), followed by site annotation and quantification (Fig. 1c). In the first step, for unambiguous site-specific phosphopeptide quantification, the phosphosite localization tool recently reported by Bekker-Jensen et al.[19] and integrated into Spectronaut was applied to filter class 1 (probability ≥ 0.75) localized sites. In the second step for site-specific quantification, an in-house program was developed to calculate the abundance of a phosphosite as the summed abundance of all the corresponding

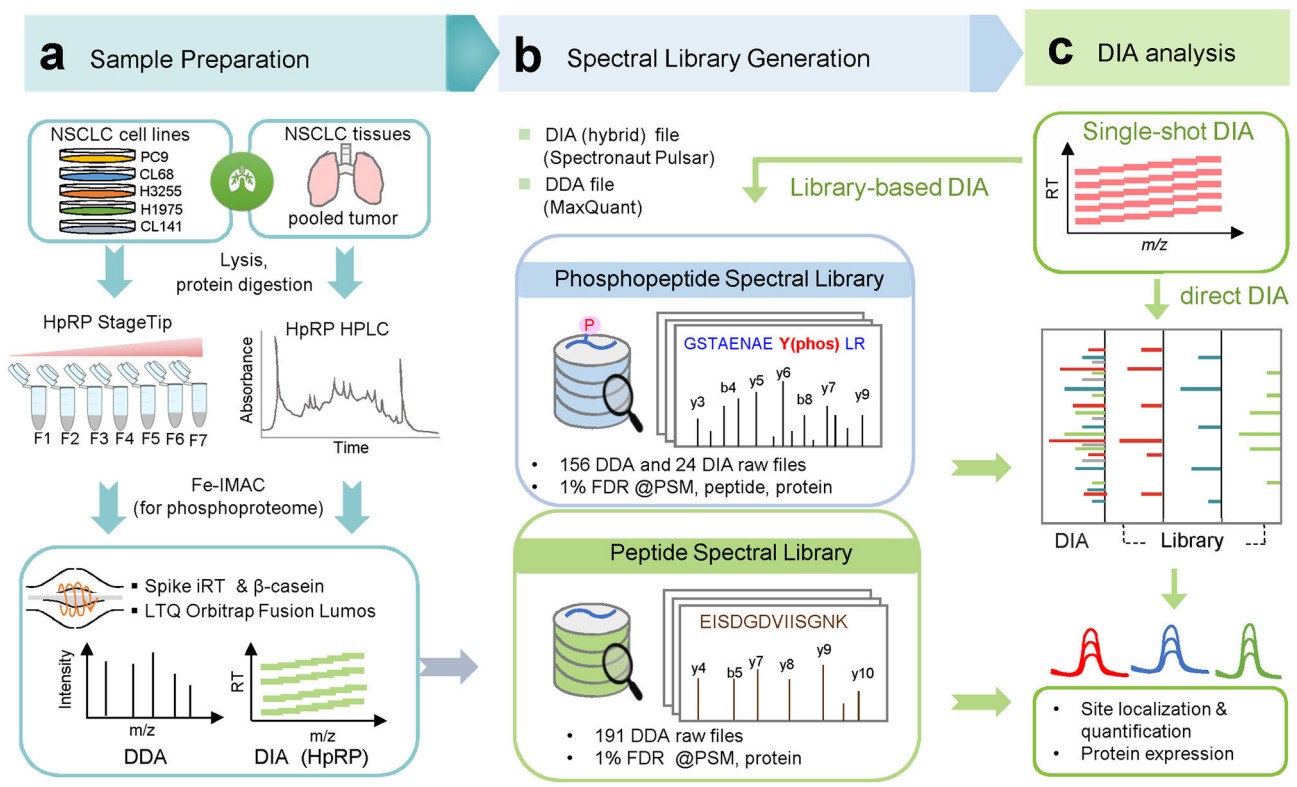

**Fig. 1 Pipeline for hybrid spectral library construction and phosphosite quantification. a** For the construction of the spectral library, non-small cell lung cancer (NSCLC) cells and tissues were lysed, digested, and fractionated by high pH reversed-phase (HpRP) chromatography in StageTip or in column (HPLC), and phosphopeptides were enriched by iron-based immobilized metal affinity chromatography (Fe-IMAC) and analyzed by the data-dependent acquisition (DDA) and data-independent acquisition (DIA) modes. The indexed retention time (iRT) standard peptides were spiked for normalization in retention time. **b** A phosphopeptide reference library was constructed from both DDA datasets ($n = 156$ raw files) and DIA datasets ($n = 24$ raw files), which were processed by Spectronaut Pulsar. Lung cancer proteome library was constructed from 191 DDA raw files processed with MaxQuant search. All data were filtered at 1% false discovery rate (FDR) at peptide spectrum match (PSM)/Precursor, Peptide, and Protein. **c** Single-shot DIA was acquired and processed by both library-based DIA (libDIA) and direct DIA (dirDIA) approach by Spectronaut. Phosphosite-level quantification was obtained by an in-house customized R program.

phosphopeptides precursor areas at the specified localization probability cutoff.

**Benchmarking quantification performance using synthetic phosphopeptides.** The phosphosite localization and quantification accuracy of the pipeline were first evaluated using 166 synthetic phosphopeptides (176 phosphosites on 109 unique sequences) selected from 61 proteins (Supplementary Data 2). The EGFR-initiated signaling cascades represent the most important pathways for lung cancer in East Asia. Thus, the synthetic phosphopeptides were selected from phosphoproteins in the lung cancer-related signaling pathways, including NSCLC signaling (61 phosphopeptides, 58 sites), EGFR-tyrosine kinase inhibitor (TKI) resistance (85 phosphopeptides, 80 phosphosites), mammalian target of rapamycin (MTOR) signaling (37 phosphopeptides, 33 sites), and PI3K-AKT signaling (73 phosphopeptides, 69 sites). In addition to 19 phosphopeptides of EGFR, other receptor tyrosine kinases (RTKs) and drug targets, including SRC, GRB2, BRAF, MTOR, MAPK1, MAPK3, MET, and EML4, were also selected to obtain synthetic phosphopeptides (Supplementary Data 2). Mono(139)-, di(21)-, and tri(6)-phosphopeptides with different sites of the same sequence were designed to evaluate the site-localization accuracy. The phosphopeptides were pooled in 5 amounts (2, 1, 0.5, 0.2, and 0.1 ng) and spiked into 0.5 μg yeast tryptic peptides. Overall, all 157 phosphopeptides (167 sites) were

in the scanning $m/z$ range of the DIA-MS method and they were all detected (Supplementary Data 2). Examples of mono- and multiple phosphopeptides of the [1161]GSHQISLDNP-DYQQDFFPK[1179] sequence from EGFR are depicted (Fig. 2a). The S1166 site can be unambiguously confirmed by b8 and b9 fragment ions, and b4 and b5 can be used to exclude phosphorylation at S1162, whereas the presence of y8 and y9 ion indicates phosphorylation at the Y1172 site. Similarly, double phosphorylation at the S1166 and Y1172 sites can be determined by a combination of the above fragment ions. In addition, distinct chromatographic elution profiles distinguish these sites (Fig. 2b). The localization probability of 222 precursors detected in the dilution series and the phosphosite localization result on the di- and tri-phosphorylated peptides with multiple competing sites at the same peptide sequence were shown in Supplementary Fig. 2. Among di-phosphorylated peptides, 18 out of 21 were confidently identified and quantified as class 1 phosphosite even in a diluted concentration, and 4 of the 6 tri-phosphorylated peptides were confidently identified and quantified. These examples demonstrate good-quality DIA spectra for unambiguous site localization.

The calibration curves for all 12 phosphosites of EGFR demonstrate good quantitative linearity ($R^2 = 0.9731$–$0.9969$) (Fig. 2c). Representative quantification curves of the autophosphorylation sites Y1197 and Y1172 in EGFR-activating mutations show good linearity ($R^2 = 0.9969$, $0.9857$) and precision (1.8–3.4% coefficient of variation, CV). Phosphosite localization

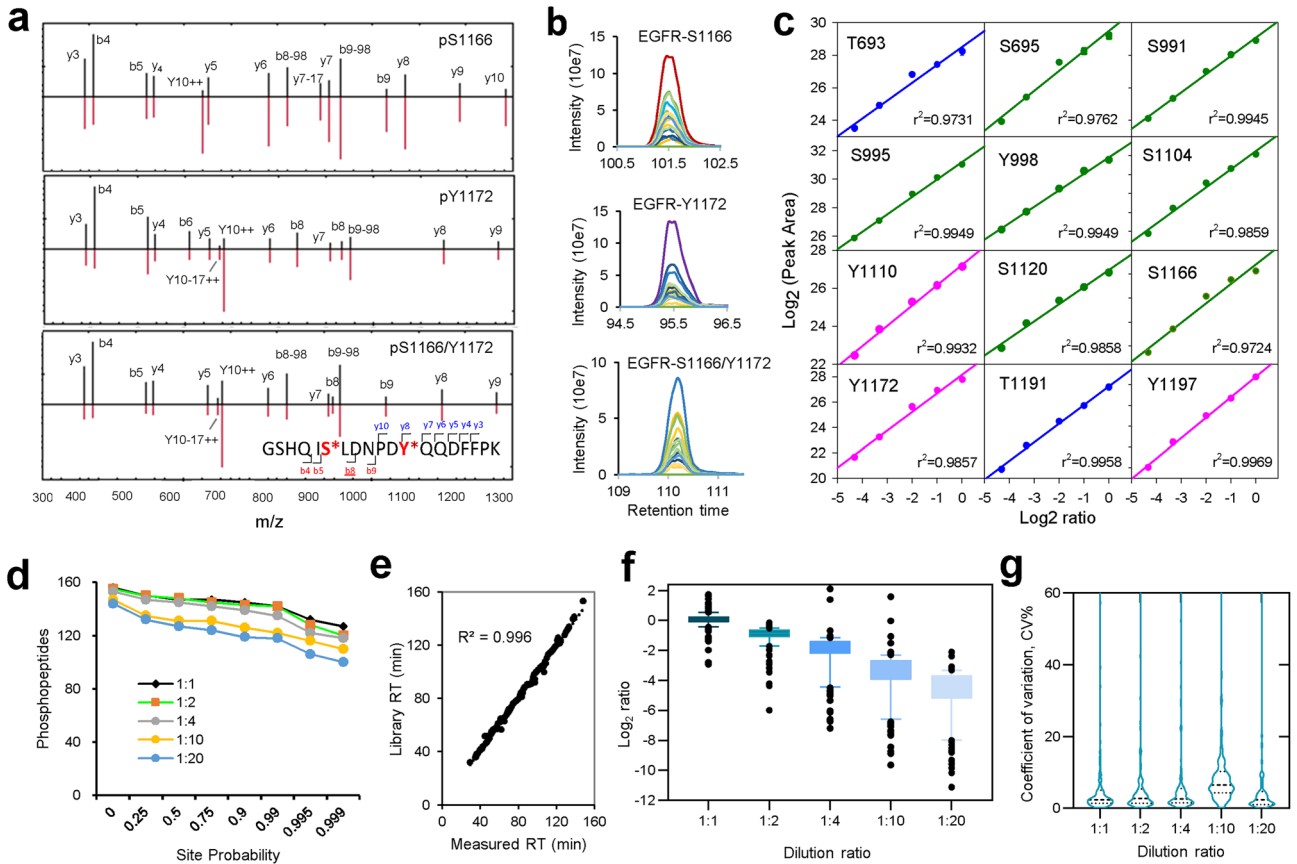

**Fig. 2 Quantification performance benchmarking using synthetic phosphopeptides.** Pooled samples of 166 synthetic phosphopeptides with serial dilutions (2, 1, 0.5, 0.2, and 0.1 ng) spiked into 0.5 μg yeast tryptic peptides was used to benchmark phosphoproteome workflow generating spectral library. **a** Example DIA spectra of mono- and multiple phosphosites on the [1161]GSHQISLDNPDYQQDFFPK[1179] sequence from EGFR. Spectra matching the DIA signal (top, black line) and fragments in the library (bottom, red line) are shown. **b** Chromatographic elution profiles of the three phosphopeptides providing unambiguous detection. **c** Quantification linearity of 12 EGFR phosphosites across dilution series using site abundance against expected theoretical dilution ratio. **d** Localization probability distribution for the quantified 157 phosphopeptides in DIA scanning $m/z$ range. Overall, among 157 phosphopeptides, 147 (93.6%), 145 (92.4%), 142 (90.4%), 131 (83.4%), and 124 (79%) were quantified as class 1 (minimum of 0.75 site-localization probability) in 1:1, 1:2, 1:4, 1:10, and 1:20 dilutions, respectively. **e** Correlation between measured retention time in DIA and expected time from library. **f** Quantification accuracy of class 1 localized phosphosites across dilution series. Box and whiskers were drawn with 10–90% percentile from $n = 3$ measurements. Median ratio of 0.95, 1.79, 3.36, 8.7, and 20.06 were observed for the theoretical ratio of 1-, 2-, 4-, 10-, and 20-fold, respectively. **g** Violin plot of distribution of coefficient of variation (CV %) of class 1 quantified phosphosites where the quartiles were shown using dot lines. A median CV% of 2.40%, 2.78%, 2.70%, 6.59%, and 2.46% were observed for phosphosites across 1:1, 1:2, 1:4, 1:10, and 1:20 dilution series, respectively. Source data are provided as a Source Data file.

analysis indicated that DIA spectra of 161 phosphosites (96%) have a high accuracy for determining the class 1 sites (probability ≥ 0.75) (Fig. 2d). With a very high stringency of 0.99 probability cutoff, 142 (90.4%), 142 (90.4%), 135 (86%), 122 (77.7%), and 118 (75.2%) were quantified from low- to high-dilution series (Fig. 2d). The measured and library-annotated retention times also showed a high correlation ($R^2 = 0.996$) (Fig. 2e). Phosphosite quantification accuracy was shown by consistency between the measured median ratios 0.95, 1.79, 3.36, 8.72, and 20.06, and expected ratios 1-, 2-, 4-, 10-, and 20-fold, respectively (Fig. 2f). The high-quantification reproducibility was evidenced by the median CV of 2.4–6.6% across the dilution series (Fig. 2g). Overall, the DIA strategy showed good spectral quality for confident site determination, high-quantification accuracy, and reproducibility.

**Construction of the phosphotyrosine-enhanced hybrid library.**
High quality and deep coverage of reference spectral libraries are crucial to achieve comprehensive profiling by DIA quantification. To construct a hybrid spectral library with complementary

coverage, tryptic peptides from different cell types and tissues were prepared by few strategies, including enhanced pTyr by PV treatment to inhibit phosphatases, peptide fractionation by reversed-phase chromatography in StageTip (cell lines) and LC (tissues) formats, and phosphopeptide enrichment by Fe-IMAC followed by analysis in DDA and DIA mode. First, DDA analyses ($n = 156$ raw files) obtained from 4 cell lines (84,368 phosphopeptides) and 6 pooled tumor tissues (39,127 phosphopeptides) and synthetic phosphopeptides (156 phosphopeptides) generated 101,624 phosphopeptides corresponding to 64,962 phosphosites. The results indicate that tumor tissues provide additional 17,817 phosphopeptides (44%) in addition to 56% commonly observed in the fractionation dataset from the cell lines, likely due to the tissue-derived or enriched proteins in tumor samples. Second, DIA analyses for 2 cell lines coupled with peptide fractionation ($n = 24$ raw files) generated 72,270 precursors (59,625 phosphopeptides), of which 26,811 (17%) were unique precursors compared to above-mentioned DDA result. Third, two cell lines, PC9 and CL68 were treated with PV enhancing tenfold Tyr phosphopeptides from 1.4% to 14% when compared to untreated

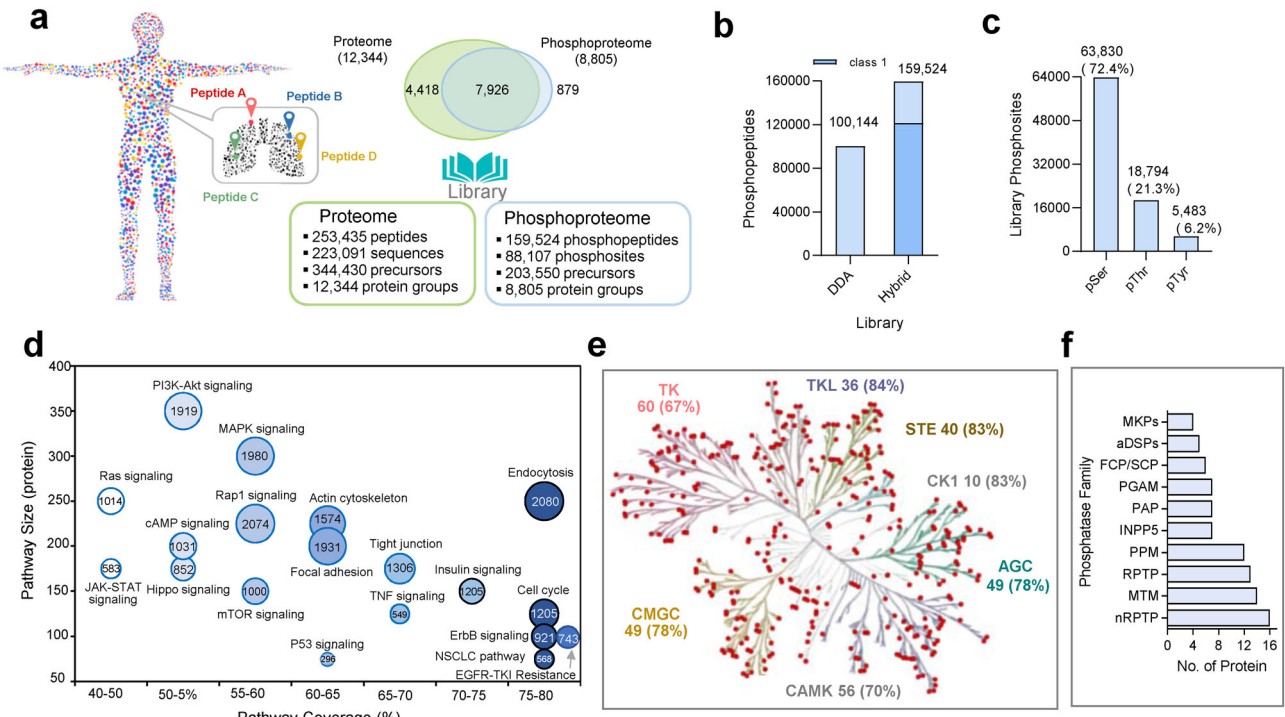

**Fig. 3 Composition of the reference spectral library for the global phosphoproteome system. a** Proteome and phosphoproteome libraries composition and protein groups overlap. **b** Phosphopeptides in hybrid and DDA-only libraries. **c** Distribution of phosphorylated serine (pSer), threonine (pThr), and tyrosine (pTyr) in the global phosphoproteome system (GPS) library. **d** Pathway annotation of phosphoproteins in the library using Kyoto Encyclopedia of Genes and Genomes (KEGG) database. The circle size corresponds to phosphosite coverage, whereas the X and Y axis shows protein coverage in the pathway and size of protein, respectively. The number in the circles shows number of phosphosites covered in the pathway. The blue color intensity shows the percentage of pathway coverage. **e** Kinase tree revealed 383 kinases in the library with proportion of each kinase family covered shown in bracket. KinMap database (http://www.kinhub.org/kinmap/) was used with input of protein accession number from GPS library. The kinase families listed includes TK (tyrosine kinases), TKL (tyrosine kinase-like), CK1 (casein kinase 1), CAMK (calcium/calmodulin-dependent protein kinase), AGC (containing PKA, PKG, PKC families), CMGC (containing CDKs, MAPK, GSK, CLK families), and STE (serine/threonine kinases many involved in MAPK kinases cascade). **f** Phosphatase obtained mapping to human dephosphorylation database (DEPOD) (www.depod.org). A total of 140 phosphatases were included in the library with the top ten families shown. The phosphatase family abbreviated are RPTPs (receptor protein tyrosine phosphatases), nRPTPs (nonreceptor-type protein tyrosine phosphatases), MKPs (MAPK phosphatases), aDSPs (Atypical DSPs), MTM (Myotubularins), PPM (protein phosphatase Mg2+ or Mn2+ dependent), FCP/SCP (TFIIF-associating component of RNA polymerase II CTD phosphatase/small CTD phosphatase), PAP (Phosphatidic acid phosphatase), INPP5 (Inositol-1,4,5-trisphosphate 5-phosphatase), and PGAM (Phosphoglycerate mutase). Source data are provided as a Source Data file.

lysates. Finally, a hybrid phosphoproteome library was generated combining fractionated DIA ($n = 24$ raw files) with the DDA ($n = 156$ raw files) dataset using Spectronaut Pulsar search with 1% FDR at PSM, peptide, and protein levels[23]. Taken together, we constructed a library consisting of 159,524 phosphopeptides (203,550 precursors corresponding to 88,107 phosphosites on 8805 protein groups; Fig. 3a and see details in Supplementary Data 3). With phosphosite localization, this library includes 121,407 class 1 phosphopeptides, (Fig. 3b), indicating that a majority have highly accurate site localization. The overall percentage of tyrosine phosphosites in the library (6%, 5483 sites of 2837 proteins) was much higher than the <1% abundance of pTyr commonly detected at the basal cellular level (Fig. 3c). Without protein immunoprecipitation, the result for EGFR phosphotyrosine sites demonstrated good coverage: nine pTyr sites were identified from the basal level of PC9 and CL68 cells bearing *EGFR* autophosphorylation. PV treatment resulted in additional sites in the kinase domain (Y764, Y978) and regulatory domain (Y1110). By similar approach, an independent protein-level spectral library of 237,701 unique peptide sequences on 12,377 protein groups was constructed from lung cancer tissues and cell lines. The phosphoproteome and proteome libraries complementarily offer identification and quantification information on the protein expression and site-specific phosphorylation.

The content of the phosphopeptide library resource was analyzed by mapping phosphoproteins to signaling pathways, the human kinome and the phosphatome. By mapping phosphoproteins to the Kyoto Encyclopedia of Genes and Genomes (KEGG) database[25], the top 29 cancer-related pathways have 42–80% coverage at the protein level with an overall 16,281 phosphosites (13,862 class 1) from 1329 protein groups. The major pathways associated with lung cancer were enriched with good coverage, including ErbB signaling (921 phosphosites, 800 class 1 on 65 proteins, 78%), NSCLC signaling (568 sites, 471 class 1 from 52 proteins, 79%), and the EGFR-TKI resistance pathway (743 sites, 648 class 1 from 59 proteins, 76%) and endocytosis (2080 sites, 1732 class 1 from 193 proteins, 80%) (Fig. 3d). In the example of the EGFR-RAS-RAF pathway, which is most crucial for lung cancer progression, 101 among 169 phosphosites are class 1 phosphosites (Supplementary Fig. 3a). Others include known kinases such as RAF1-S642, BRAF-S365, and BRAF-T753, which are known to be phosphorylated by ERK2 kinase, as well as MEK1/MEK2 substrate site (ERK1-T202, ERK1-T204, ERK2-T185, and ERK2-Y187) and RAF substrate site (MEK2-S222). In addition, EGFR-S1036, SOS1-S232, and SOS1-1205 are among the newly identified class 1 sites. For the oncogene *EGFR*, the 55 phosphosites (40 class 1 sites) located in the tyrosine kinases and autophosphorylation domain also

include the Y1197 and Y1172 phosphosites, which are the characteristic autophosphorylation upon activating driver mutation of *EGFR* (Supplementary Fig. 3b). By mapping 522 human kinases deposited in KinMap[26] and 238 human phosphatases deposited in DEPOD[27], 383 kinases (73%) with 5091 phosphosites (4268 class 1) in the kinase tree and 140 (59%) phosphatases with 1429 phosphosites (1,226 class 1) were covered in our library including major groups (Fig. 3e, f).

The 88,107 phosphosites in this library provide deep phosphoproteome coverage from a single cancer type over a single MS platform. Compared to the phosphosite coverage from single cell type (HeLa) of 50,497 phosphosites reported by Sharma et al.[8], our DDA data (64,962 sites) in the GPS library processed by the same search platform of MaxQuant still presented 28.6% increase of phosphosites (Supplementary Fig. 4a). Compared to the PhosphoSitePlus[28] database (239,180 phosphosites, as of 23 October 2020), our library contains 26,234 additional phosphosites (17,097 sites from hybrid class 1 localized library), including 1589 pTyr sites (Supplementary Fig. 4b and Supplementary Data 3). Examples include novel sites with high localization probability ≥ 0.95: EGFR-S1036, STAT3-T716, SRC-S212, ALK-S76/77/78, and PLCG2-Y13/S785, in the NSCLC pathway. The "plug-and-play" database (109,611 phosphosites, 11,428 proteins) was constructed on the datasets of 989 LC-MS/MS runs using phosphopeptides enriched with Fe-IMAC and TiO$_2$ from MCF7, HeLa S3, and HepG2 cell lines over different MS instruments (LTQ Orbitrap Velos, Elite, Fusion, and Q-Exactive), which is a comprehensive resource to provide targeted assays in phosphoproteome analysis[15]. Compared to the "plug-and-play" database, 73,808 (46% of 159,524) phosphopeptides were in common with 85,716 phosphopeptides that are uniquely present in our GPS library (Supplementary Fig. 4c), suggesting the complementary nature of our GPS library.

**Lung cancer proteome spectral library**. Interpretation of quantitative phosphoproteomics data require considering changes in both protein expression and phosphorylation degree[29]. Large-scale human proteome libraries have been recently reported over different MS platform as a valuable tool[30,31]. In addition to a phosphoproteome library, we constructed a protein-level library in parallel from 5 NSCLC cell lines and 22 pooled tumor tissue samples. Peptide fractionation was performed for tryptic peptides from cell lines using reversed-phase chromatography in StageTip format, whereas reversed-phase column chromatography was used for tissues. Using the same chromatographic and MS instrument, a total of 191 raw files were generated in DDA mode (Supplementary Fig. 5). MaxQuant-based protein identification was performed at 1% FDR of PSM and protein level. Finally, a protein-level spectral library of 344,430 peptide precursors of 223,091 unique peptide sequences on 12,344 protein groups was constructed.

Quantification performance evaluation using triplicate analysis of PC9 and CL68 cell lines against the proteome library achieved protein groups of 7618 in PC9 and 7793 in CL68 (Supplementary Fig. 6). Compared to DDA, a 1.4-fold more quantified proteins were achieved with much lower missing values and high reproducibility The phosphoproteome and proteome libraries offer deep coverage and complementary identification and quantification information on the protein expression and site-specific phosphorylation level.

The performance of a hybrid library was evaluated by comparing the spectra library constructed from a single acquisition type of DDA or DIA. Two datasets of comparable size were generated from DDA and DIA using three cell lysate digests coupled with StageTip fractionation: proteome from HeLa and phosphoproteome from PC9 and CL68 cell lines (Supplementary Fig. 7a). Three sets of proteome and phosphoproteome spectra libraries were independently constructed by using DDA, DIA, and hybrid datasets of equal numbers of raw files. For the library construction from HeLa lysate ($n = 16$), the number of peptides in the hybrid library increased by 8% and 22% compared to DDA and DIA, respectively, showing the expected complementary nature of merging DDA and DIA datasets in the hybrid library (Supplementary Fig. 7b). By using another triplicate runs of single-shot DIA dataset from HeLa cells, the hybrid library achieved the highest coverage with 12% and 5% more quantified peptides compared to the DDA-based and DIA-based libraries, respectively, whereas the DIA-based library outperformed the DDA-based library (Supplementary Fig. 7b). Similarly, for phosphoproteome dataset from PC9 and CL68, the hybrid libraries ($n = 14$) also resulted in the highest number of phosphopeptides (Supplementary Fig. 7c). Besides, the single-shot DIA data mapping to the DIA-based library still outperformed the result from the DDA-based library likely due to the similar nature of fragmentation pattern. Nevertheless, quantitative comparison showed high correlation ($R^2 > 0.9$) for the quantified phosphopeptides among the three libraries (Supplementary Fig. 7d).

**Single-shot DIA offers highly reproducible large-scale phosphosite quantification**. Using NSCLC PC9 and CL68 cell lysates, we next evaluated the quantification performance in phosphoproteome coverage, CV%, and data completeness of single-shot triplicate DIA analysis. The single-shot dirDIA and libDIA results were obtained from Spectronaut and were compared to the results from single-shot DDA and peptide-fractionated DDA data processed by MaxQuant label-free quantification (LFQ) with 1% FDR at site, PSM, and protein level (Fig. 4a, b). In the 2 h gradient LC-MS/MS, the single-shot DDA analysis in PC9 cells quantified 17,430 phosphopeptides corresponding to 12,959 phosphosites (10,768 class 1 sites), yet only 53% sites were reproducibly quantified within a 20% CV among triplicates. Peptide fractionation using HpRP StageTip prior to DDA analysis (7 fractions in duplicate) enhanced the quantification to 36,676 phosphopeptides corresponding to 23,527 phosphosites (18,631 class 1 sites) (Supplementary Data 4). By the same 2 h gradient, the dirDIA analysis achieved 19,024 phosphosites (9746 class 1 sites) and as high as 90% were quantified within a 20% CV. By using our constructed library, the libDIA approach achieved 33,330 phosphosites (17,810 class 1 sites) and 79% were quantified within a 20% CV, which were 1.9- and 1.6-fold more class 1 phosphosites than using DDA and dirDIA, respectively (Supplementary Data 5). The complementary libDIA and dirDIA approaches quantified 34,886 and 38,255 phosphosites for PC9 and CL68, respectively, with 95% unique phosphosites covered by libDIA (Fig. 4b). The DIA-based quantification resulted in enhanced phosphosite coverage with 1.4-fold more quantified sites than that in the HpRP fractionation using the DDA method (28 h LC-MS/MS). The superior quantification of libDIA is likely due to the more efficient detection of low-intensity peptide ions in the DIA mode as well as the high coverage of pTyr sites in our library.

In this study, we constructed a reference library from lung cancer samples and further demonstrated enhanced profiling coverage and quantification performance. Given the high percentage of similar proteome composition within human tissue, the general utility of GPS for diverse sample types was evaluated using other human specimens from breast cancer. Using 200 µg MDA-MB-453 breast cancer cell from DIA analysis over a 2 h gradient LC-MS/MS, the libDIA (41,014 phosphopeptides) still outperformed the dirDIA (26,868 phosphopeptides) with a

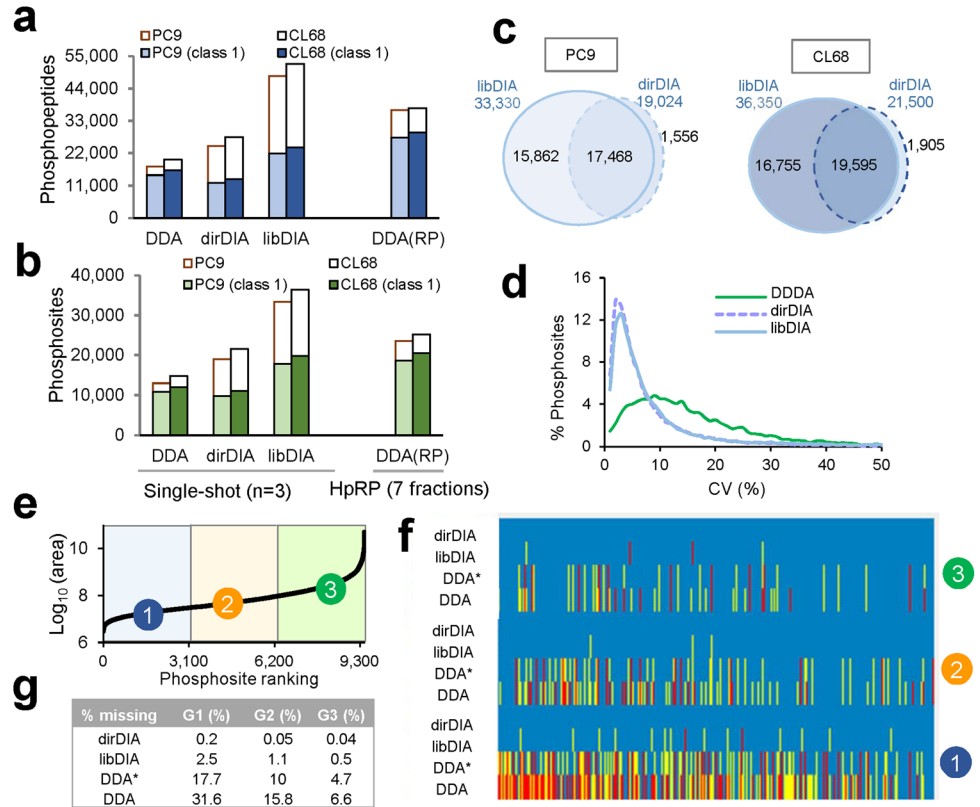

**Fig. 4 Comparison of quantification performance in DDA and DIA using cell lysate.** Two NSCLC cell lysate of PC9 and CL68 were processed by both DDA and DIA mode including reverse-phase (RP) fractionation. The DDA data were searched by MaxQuant, whereas DIA was analyzed by Spectronaut in both library-based and direct DIA mode. **a** Summary of phosphopeptides identified by single-shot DDA, StageTip fractionated DDA (7 fractions run in duplicate), library-based DIA (libDIA), and direct DIA (dirDIA). **b** Phosphosite identification comparison. The single-shot DDA and DIA were acquired in triplicate. **c** Overlap of phosphosites between dirDIA and libDIA. **d** Distribution coefficient of variation, CV% of phosphosites of PC9 cell ($n = 9,665$ for DDA, 19,007 for dirDIA, and 30,260 for libDIA. A median coefficient of variation (CV) value of 13.0%, 4.3%, and 5.2% were obtained for DDA, dirDIA, and libDIA in PC9, respectively. **e** Distribution of phosphosite abundance rank of commonly quantified 9456 phosphosites in DDA, DDA match between runs (represented as DDA*), libDIA, and dirDIA. **f** Phosphosites identification per each abundance group across triplicate measurements. The blue, yellow, and red lines represent sites quantified in all the three, two, or only in one replicate, respectively. **g** Missing values across different abundance groups. Source data are provided as a Source Data file.

median quantification CV% of only 4% in both cases (Supplementary Fig. 8). The result is comparable to the lung cancer cell result with slightly lower coverage, likely due to the absence of breast cancer-related proteins in the library. Compared to the reported phosphoproteomic profiling of breast cancer cell[32] (~14,000 class 1 phosphosites) over 270 min reversed-phase peptide separation by single-shot DDA, the result demonstrated highly sensitive quantification of cross-cancer phosphoproteomic profiling by our GPS strategy.

Compared to the distributions with a median CV value of 13.9% in the DDA, the libDIA and dirDIA results had much narrower distributions with median CV values of 5.2% and 4.3%, respectively, showing a significantly higher quantification reproducibility in DIA (Fig. 4d). In particular, libDIA shows superior quantification of class 1 tyrosine phosphosites (721 sites,) compared to dirDIA (242 sites), whereas only 93 sites were quantified by DDA in PC9 cells (Supplementary Fig. 9). Furthermore, 10–50% between-run missing values are commonly observed in LFQ[33], presenting a bottleneck for reproducible quantification across large numbers of samples. To evaluate the between-run missing values, 9456 commonly quantified phosphosites were grouped into three categories according to their abundance rank (Fig. 4e). In the most abundant group (G3), DDA showed <6.6% missing values among triplicate runs, whereas both libDIA and dirDIA showed 0.5% and 0.04%

missing values, respectively. In the lowest abundance region (G1), significantly low between-run missing values of 2.5% and 0.2% were observed in libDIA and dirDIA, respectively, while DDA reached 31.6% and improved to 17.7% by using the match between run (MBR) feature (Fig. 4f, g). These results highlighted the advantages of DIA to allow deep profiling and highly reproducible quantification between runs, which are critical benefits for multiplexed quantification, such as clinical proteomics, for many specimens.

**Differential phosphoproteomics profiling of EGFR-TKI-sensitive and EGFR-TKI-resistant lung cancer cells.** Despite the efficacy of targeted therapy using TKI for patients with activating driver mutation (two major types: Del19 and L858R) on EGFR, management of these patients who eventually develop resistance to EGFR-TKI has become the biggest challenge in lung cancer therapy and remains unmet clinical need[10,34,35]. In East Asia, the primary cause of resistance to TKI is driven, in ~60% of advanced lung cancer patients, by acquiring an additional EGFR T790M point mutation located at the gatekeeper position of the adenosine triphosphate-binding site[36]. We applied the GPS approach to quantitatively compare TKI-sensitive PC9 with exon-19 deletion ($IC_{50} = 30$ nM) and TKI-resistant CL68 ($IC_{50} = 20$ μM) cells with a double mutation of exon-19 deletion and T790M point mutation after Iressa and chemotherapy treatment,

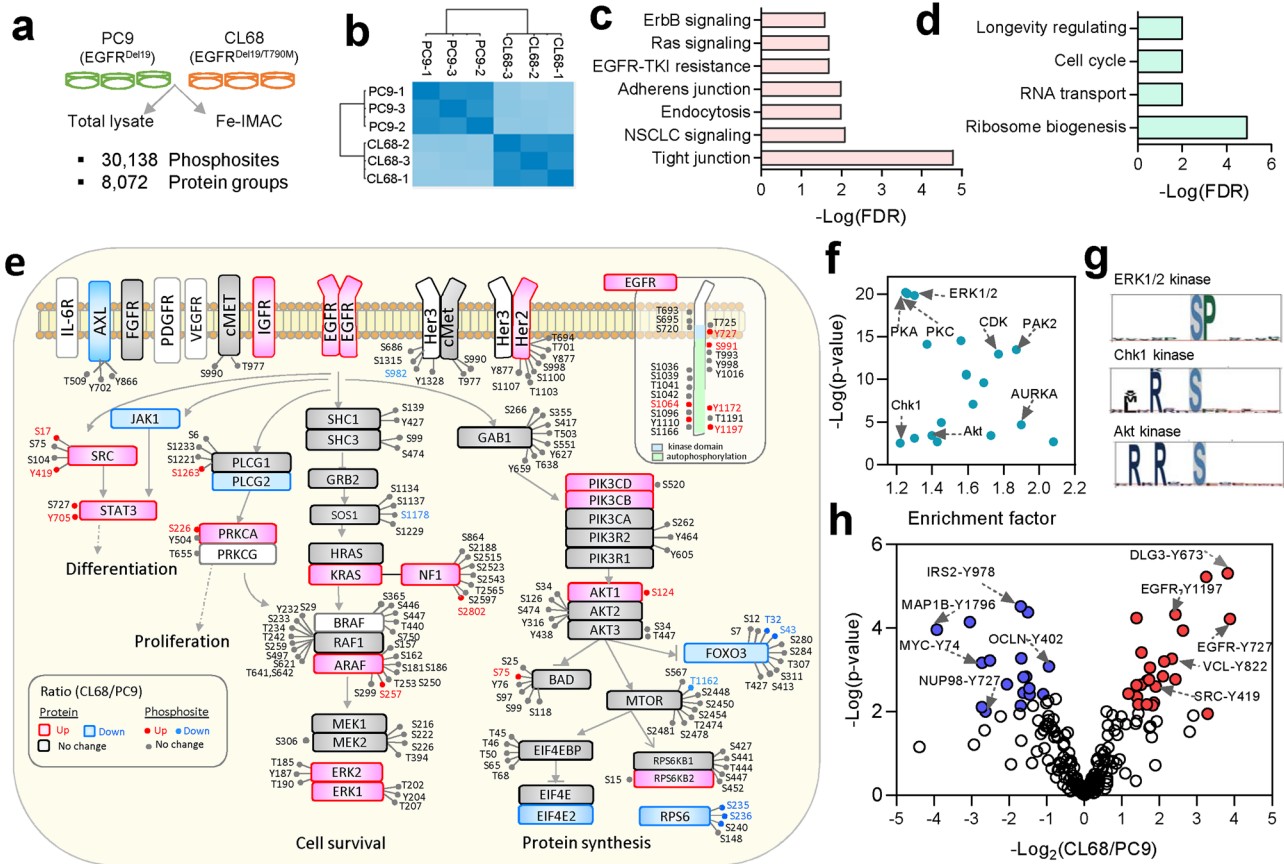

**Fig. 5 Differential phosphoproteome profiling of EGFR-TKI-sensitive and resistant lung cancer cells. a** Experimental design and summary of the identification and quantification results of the proteome and phosphoproteome in EGFR-TKI-sensitive (PC9) and EGFR-TKI-resistant (CL68) lung cancer cells in three biological triplicates. **b** Pearson's correlation of phosphosite abundance in biological triplicate analysis. KEGG signaling pathways enriched ($p < 0.05$) from **c** upregulated and **d** downregulated phosphosites/phosphoproteins using STRING (two-sampled $t$-test, FDR < 0.01, S0 = 0.1). **e** Overall, 161 phosphosites and 49 proteins were quantified in the EGFR-TKI resistance signaling pathway, revealing differentially expressed proteins (two-sampled $t$-test, FDR < 0.01, S0 = 0.4) and phosphosites (two-sampled $t$-test, FDR < 0.01, S0 = 0.1). **f** Kinase motfi enrichment analysis using Fisher's exact test (FDR < 0.02) where top ones shown among 30 kinase motifs. **g** Motif logo for ERK1, 2, and PKA or PKC kinases extracted by pLOGO (https://plogo.uconn.edu/). **h** Tyrosine phosphosites showing differential expression levels between the two cell lines (two-sampled $t$-test, FDR < 0.01, S0 = 0.1). Source data are provided as a Source Data file.

which may provide insight into the drug resistance mechanism[37]. A total of 16,199 class 1 phosphosites on 4122 proteins were quantified (Fig. 5a and Supplementary Data 6). High correlation was observed (Pearson's correlation = 0.95) from three biological replicas, indicating high reproducibility of quantitative phosphoproteomic results (Fig. 5b). At the protein level, the expression of 82% phosphoproteins were also quantified.

Differential expression of the quantified phosphosites, resulted in 747 upregulated and 1011 downregulated phosphosites in resistant cells compared to phosphosites in sensitive cells (two-sample $t$-test, FDR < 0.01, S0 = 0.1) Supplementary Data 6). Pathway analysis of upregulated phosphoproteins against KEGG database enriched the top ranking pathways, including NSCLC signaling, ErbB signaling, Ras signaling, endocytosis, and the EGFR-TKI resistance pathway, which have been reported to be associated with TKI resistance in NSCLC (Fig. 5c). Several cancer-associated pathways, such as adherens junctions, tight junctions, and focal adhesions associated with epithelial–mesenchymal transition (EMT), were also enriched (Fig. 5c). EMT has been reported as a major hallmark of EGFR-TKI resistance in NSCLC[38]. Our results may reveal elevated site-specific phosphorylation in an EMT event. Among downregulated phosphosites, RNA transport and ribosome biogenesis-related pathways are enriched (Fig. 5d). High coverage and alterations in phosphosite and protein expression were observed in mapped

pathways. Among the most deregulated pathways ($p < 0.05$), e.g., the EGFR-TKI resistance pathway, 161 phosphosites covering almost all downstream proteins were observed and 49 out of 78 proteins were quantified at the proteome level (Fig. 5e). Twenty phosphosites showed differential levels, such as the higher phosphorylation level of a known autophosphorylation site (Y1197 and Y1172) and kinase domain (Y727) on EGFR, accompanied by protein overexpression at the PI3K/Akt and SRC/STAT3 subpathways.

To explore the upstream kinases responsible for TKI resistance, kinase motif enrichment of differentially expressed phosphosites was performed. Motif enrichment (Fisher's exact test, FDR < 0.02) identified 30 motifs with prominent roles in serine and threonine kinases, including top ranking protein kinase A (903 substrates), protein kinase C (858 substrates), ERK1/2 kinase motif (755 substrates) (Fig. 5f, g), as well as tyrosine kinases Src with their overexpressed substrates ($n = 18$) in the TKI-resistant CL68 cells (Supplementary Data 6). In addition, upregulation of the TPX2-S121 and S125 sites likely correlated with the reported role of AURKA kinase and its coactivator TPX2 in response to chronic EGFR inhibition to mitigate drug-induced apoptosis in resistant lung cancer cells[39]. The checkpoint kinase 1 (Chk1) is among kinases enriched with known phosphorylation site using PhosphositePlus (Fisher's exact test FDR < 0.02). Among

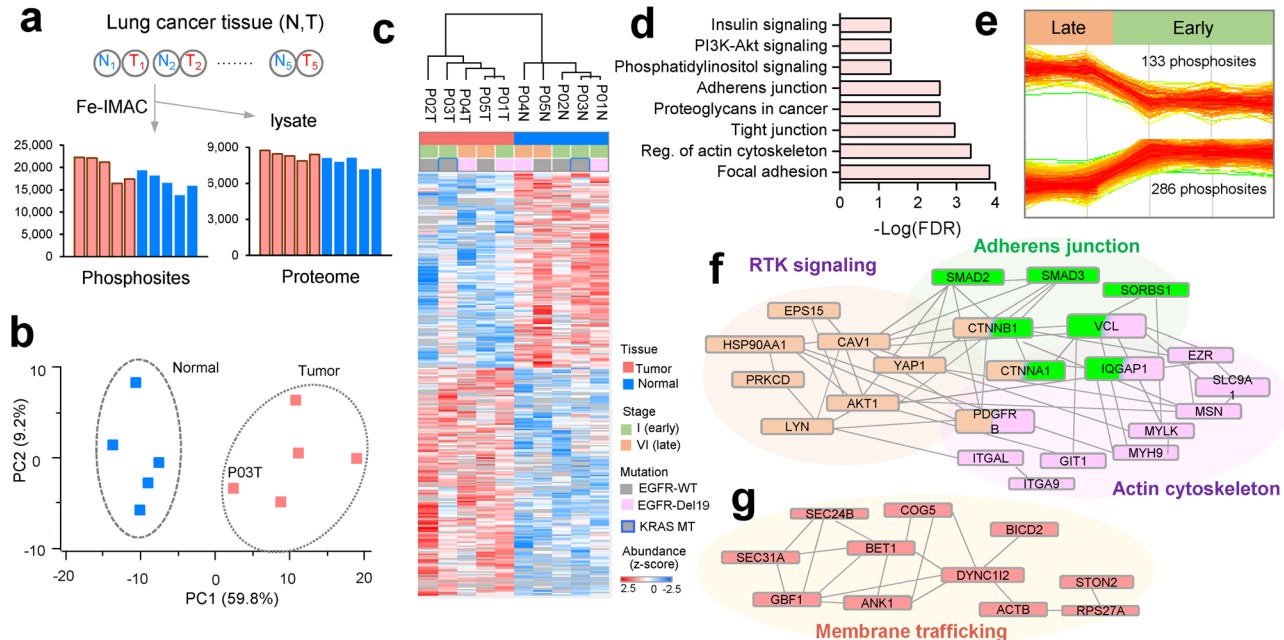

**Fig. 6 Phosphoproteome profiling of lung cancer tissues. a** Summary of phosphosite and protein-level identification results in each tumor (T) and adjacent normal (N) lung cancer tissue. **b** Principal component (PC) analysis of tumor and adjacent normal tissues using differentially expressed sites normalized to protein level (two-sampled *t*-test, $p < 0.05$, S0 = 0.1). **c** Unsupervised clustering of 585 (normalized) differentially expressed phosphosites between tumor and normal tissues (two-sampled *t*-test, $p < 0.05$, S0 = 0.1). **d** Pathway analysis of upregulated phosphosites by the Kyoto Encyclopedia of Genes and Genomes (KEGG) database (FDR < 0.05). **e** Differentially expressed phosphosites showing increasing or decreasing expression between early and late stages (two-sampled *t*-test, $p < 0.05$, S0 = 0.1). Protein–protein interaction network and functional category of phosphoproteins for **f** upregulated in the early and **g** upregulated in late stages filtered with medium confidence and FDR < 0.05 in STRING database (https://string-db.org/). Source data are provided as a Source Data file.

upregulated phosphosites, MATR3-T150, EML3-S176, ERRFI1-S302, LMO7-S1510, and TRIM28-S473 are its known substrates. Chk1 has been reported to be associated with tumor proliferation[40] and is a resistance drug target with ongoing clinical trials[41]. We further quantitatively compared the alterations in 646 pTyr sites, of which 43 sites showed differential phosphorylation (Fig. 5h). Many upregulated sites are associated with EGFR (Y1197, Y727) and its adaptor proteins GAB1-Y627 and DLG3-Y673, likely due to EGFR-activating mutations that drive its downstream signaling cascade. Other sites include pY62/63 of the tyrosine phosphatase Shp2; Shp2 knockdown has been reported to increase cellular sensitivity to gefitinib in EGFR-TKI-resistant lung cancer cells[42]. Whether these identified kinases and phosphosites may confer the transformation from TKI-tolerant to TKI-resistant cells remains to be validated.

**Deep phosphoproteome profiling in lung cancer tissues by DIA-MS.** We further applied single-shot DIA analysis for proteome and phosphoproteome profiling in paired tumor and adjacent normal tissues from five lung cancer patients in early and late stages (Supplementary Data 1). Phosphoproteomic analysis resulted in 32,407 phosphosites (18,417 class 1) on 4777 proteins (Fig. 6a). The proteome analysis quantified 9294 proteins using proteome library. Overall, 16,103 class 1 phosphosites were commonly quantified at both the protein and phosphosite levels. After normalizing the phosphosite abundance with protein expression, 585 phosphosites on 446 proteins showed differential expression (two-sample *t*-test, S0 = 0.1, $p < 0.05$) between tumor and normal tissues (Supplementary Data 7).

Principal component analysis of the differential phosphosites revealed distinct expression between tumor and normal tissues,

and high heterogeneity of the tumor profiles (Fig. 6b). Unsupervised clustering also revealed distinct profiles with differentially expressed phosphosites between tumor tissues than from adjacent normal tissues (Fig. 6c). Pathway analysis of upregulated sites (phosphoprotein) using the KEGG database in STRING enriched the top pathways related to focal adhesion, regulation of actin cytoskeleton, tight junction, as well as PI3K-Akt signaling and insulin signaling ($p < 0.05$) (Fig. 6d). These upregulated sites included two clinical drug targets annotated as Src-family kinases (HCK and LYN), which were reported to promote tumor malignancy in NSCLC[43]. Interestingly, PI3K-Akt signaling pathway well known to promote oncogenesis in lung cancer was activated including phosphosites of PTEN, ITGB4, PTK2, RPTOR, and others[44]. Mitogen-activated protein kinase pathway-related phosphoproteins including MAP3K4, PRKCA, DUSP16, FAS, and others were also upregulated in tumors. Phosphosites from immune system-associated proteins (e.g., IL3RA, PECAM1, PTPN1, SIGLEC7, and JUP) showed an overall higher phosphorylation in tumor tissues than in normal adjacent tissues (Supplementary Data 7), suggesting enhanced immune signaling is likely to occur during tumor development.

We further evaluated if distinct phosphosites can associate with the progression from three early-stage tumors (IA, IB) to two late-stage (IV) tumors. Overall, significantly more phosphosites were upregulated in late-stage tumors than in early-stage tumors ($p < 0.05$, two-sample *t*-test) (Fig. 6e). Further network analysis ($p < 0.05$) revealed higher levels of actin cytoskeleton- and adherens junction-related phosphoproteins in early-stage tumors than in late-stage tumors (Fig. 6f), showing the reorganization of the cell membrane in the early phase of tumor progression. Signaling by RTKs was also found to be deregulated in early-stage tumors. These include AKT1-S124, CAV1-S37, LYN-S11,

PDGFRB-S705, EPS15-S814, EPS15-S814, PRKCD-S645, and ITGAL-S1140 phosphosites. The phosphosites associated with membrane trafficking showed an increasing trend in late-stage tumors (Fig. 6g). The deep tissue profiling revealed the differential site-specific phosphorylation and may provide further insights to suggest potential drug targets.

To evaluate the technical advancement on the phosphoproteomics profiling for large-scale analysis of clinical samples by the GPS, the above tissue phosphoproteomics profiling results were compared to our previous large-scale tandem mass tag (TMT) phosphoproteomic datasets of tissues samples from NSCLC patients[10]. From combined datasets of tissue samples, <5% of 166,792 phosphopeptides were reproducibly quantified in at least 75% of the samples (0.2 mg per tissue) in 80 patients; a large amount of phosphoproteomics TMT data were generated but limited number overlap across all samples. As a comparison, reproducible profiling of >30,000 phosphosites were achieved for the 5 pairs of tissues without peptide fractionation strategy. Although comparison of TMT with DIA was not the main scope of this study, the comparison also demonstrated superior quantification of DIA to improve the known problem of ratio compression (tumor-to-normal). By comparing the phosphosites ratio for five pairs of tissues from DIA and the TMT datasets (Supplementary Table 1), the DIA results have generally larger ratios compared to those in the TMT dataset. For example, EGFR-T693 known to have decreased expression in lung cancer has more obvious downregulation in the DIA result. In summary, a DIA-based, label-free approach may offer an efficient alternative for large-scale phosphoproteomic profiling of tissue samples requiring much lower starting amount.

## Discussion

Compared to the conventional spectral libDIA approach, a recent study showed that the library-free approach (dirDIA) offers the advantage of ease of application with quantification coverage of ~20,000 phosphopeptides (9500 localized phosphosites[19]). Our results showed that libDIA significantly outperforms all methods and the integrated dirDIA and libDIA pipeline reveals a complementary profiling result likely due to different data deconvolution. On the demonstration from a cell line to human tissue, we reported integrated single-shot DIA for fast (2 h gradient), highly reproducible and large-scale phosphoproteome profiling (36,350 quantified phosphosites) with comparable coverage to the fractionation approach (25,163 phosphosites, 7 fractions). Our result of 95% unique phosphosites covered by libDIA-based identification and quantification revealed the strength of the targeted approach using the rich fragment peaks to map large-scale reference libraries with site-specific localization accuracy. We believe that the integrated DIA pipeline combined with a comprehensive library will advance phosphoproteomics applications to diverse samples. Although the complementary identification and quantification results of libDIA and dirDIA approaches present a potential opportunity to increase the proteome and phosphoproteome-profiling depth, merging the output from libDIA and dirDIA approaches or different software tools will require future development of dedicated FDR control strategies to ensure the identification confidence.

In the current human phosphoproteome database PhosphoSitePlus, ~240,000 Ser, Thr, and Tyr residue phosphosites were deposited. Using lung cancer samples as a model, our hybrid phosphoproteome reference library of over 88,107 sites contains 26,234 newly identified phosphosites compared to existing public repositories, suggesting that the phosphoproteome is likely significantly underexplored in a sample-specific manner. In particular, the observed high number of tyrosine sites in the

single-shot DIA results that outperform the fractionation-based DDA results is likely attributed to the enhanced coverage of the tyrosine sites in the reference library (6%). The established proteome spectral library also contains 12,344 proteins (223,091 peptide sequences) for complementary protein expression analysis. With the demonstrated good pathway coverage, such experimentally verified spectral resources with novel sites can be useful as a reference DIA digital map for the targeted monitoring of signaling pathways. These verified spectra can also be a training dataset to expand our knowledge in machine learning-based spectral prediction of undiscovered phosphorylation[45]. DIA achieved sensitive and highly reproducible profiling of an EGFR-TKI-resistant cell model at both the proteome and phosphoproteome levels, uncovering a high coverage of the EGFR-TKI resistance pathway and differentiating alterations in the expression and site-specific phosphorylation of novel kinases and substrates. The highly consistent depth and reliable quantification across the highly heterogeneous tumor and adjacent normal tissue of all patients revealed the power of DIA to understand disease mechanisms and mine potential drug targets. Our GPS spectral library is currently limited to lung cancer samples and will have to be extended to other cancer and sample types, to make it a more generic resource for DIA analysis. Besides, application to large-scale clinical samples to derive insight into disease mechanism remains to be evaluated. Further advancement in DIA strategy, such as more efficient DIA acquisition modes over different MS platforms, and extending spectral libraries applicable to multiple cancer types will further enhance applicability of the technology. Besides, informatics tools for error rate estimation to integrate targeted and dirDIA analysis results will further improve coverage and quantification accuracy to implement the strategy towards clinical application. With the cancer spectral library made freely available, we believe that such a DIA digital map and integrated single-shot DIA strategy will advance phosphoproteomics applications for diverse sample types.

## Methods

**Chemicals and materials**. Formic acid (FA), chloroform, sodium laurate (sodium dodecanoate, SL), sodium deoxycholate (SDC), sodium lauroyl sarcosinate (SLS), dithiothreitol (DTT), iodoacetamide (IAM), triethylammonium bicarbonate (TEABC), tris (2-carboxyethyl) phosphine hydrochloride (TCEP), 2-chloroacetamide (CAA), ammonium hydroxide, phosphatase inhibitor cocktail 2, and phosphatase inhibitor cocktail 3 were purchased from Sigma Aldrich (St. Louis, MO, USA). Ammonium formate was obtained from Fluka. Trifluoroacetic acid (TFA), methanol, ethyl acetate, and lysyl endopeptidase were purchased from WAKO (Osaka, Japan). Acetonitrile (ACN) and acetic acid were purchased from Merck (Bedford, MA, USA). Modified sequencing-grade trypsin and yeast protein extract were purchased from Promega (Madison, WI, USA). SDB-XC Empore™ disks, Styrene Divinylbenzene-reverse-phase (SDB-RPS) Empore™ disks, and C8 membrane were purchased from 3M™ (St. Paul, MN, USA). Ni-NTA silica resin was purchased from Qiagen (Hilden, Germany). Iron (III) chloride (FeCl₃), acetic acid, high-performance liquid chromatography (HPLC)-grade ACN and FA, and BCA™ protein assay kit were purchased from Pierce (Rockford, IL, USA). Five-micrometer C18-AQ beads were purchased from Dr. Maisch-GmbH (Ammerbuch, Germany). Synthetic phosphopeptide standards were purchased from Synpeptide Co. Ltd (Shanghai, China). The iRT peptide kit was purchased from Biognosys AG (Schlieren-Zurich, Switzerland). Water was obtained from a Millipore Milli-Q System (Millipore, Bedford, MA, USA).

**Cell culture and lysis**. The human lung adenocarcinoma cell line PC9, CL68, H3255, CL141, and H1975 were gifts from Dr P.C. Yang (Department of Internal Medicine, National Taiwan University). The detailed source and clinical information of the cell lines were shown in Supplementary Data 1. The MDA-MB-231 breast cancer cell line was purchased from Bioresource Collection and Research Center, Taiwan. The cell lines were grown in RPMI-1640 medium containing 10% fetal bovine serum, 2 mM ʟ-glutamine (Life Technologies, Inc.) and 1% penicillin G (GibcoBRL, Gaithersburg, MD, USA) at 37 °C in a humidified atmosphere of 5% CO₂/95% air. For experiments to enhance the number of tyrosine phosphosites, the PC9 and CL68 cells were treated with 250 μM PV (pH 10, with 0.14% H₂O₂) for 40 min before collection. Cells were washed three times with phosphate-

buffered saline (PBS, 0.01 M sodium phosphate, 0.14 M NaCl pH 7.4) (Sigma, St. Louis, MO, USA) and collected in lysis buffer cocktail (1% SL buffer, 10 mM TCEP, 40 mM CAA, protease inhibitor, and phosphatase inhibitors in 100 mM Tris pH 8.5). The collected cells were heated to 95 °C for 5 min and sonicated at 4 °C for 30 min. The lysate was then centrifuged at 16,000 × g for 20 min at 4 °C. The supernatant was collected and subjected to StageTip-based digestion.

**Lung cancer tissue collection and lysis.** Clinical tissues from lung cancer patients were obtained from National Taiwan University Hospital confirmed by pathologist. All ethical regulations have been complied and approved by Institutional Review Board on Biomedical research of Academia Sinica and National Taiwan University Hospital Research Ethics Committee. All patients have provided written informed consent. Following surgery, the tumor and adjacent normal tissues were collected from the most distal relative site in separate tubes, kept on dry ice for 30 min during transportation, and stored at −80 °C before further processing. In this study, 25 tissue samples were collected and analyzed either for construction of spectra library or DIA-based quantification of individual patients. The clinical information of lung cancer patients is shown in Supplementary Data 1.

Tissue samples were processed according to our recent report with some modifications[10]. Frozen tissues were thawed rapidly on ice, cut into small pieces, weighed, and then washed by an ice-cold PBS buffer to remove blood. The pre-cleaned tissues were homogenized in tenfold volume of lysis buffer solution containing 12 mM SDC, 12 mM SLS, 100 mM Tris-HCl pH 9.0, phosphatase cocktail inhibitors, and EDTA-free protease cocktail inhibitor under 4 °C using mechanical homogenizer (Precellys®24, Bertin Technologies). The homogenized samples were heated at 95 °C with vortexing at 750 r.p.m. for 5 min to inactivate the endogenous proteases and phosphatases[4], and sonicated for 10 min (30 s on, 30 s off) using Bioruptor Plus (Diagenode, Denville, NJ). Residual debris was removed by centrifugation (16,000 × g for 30 min at 4 °C) and the supernatant was collected. The protein concentrations were determined via BCA protein assays.

**Protein digestion.** For cell line samples' phosphoproteome analysis, aliquots of cell lysates were digested in a StageTip using our recently developed streamlined approach modified from Kulak et al.[46]. Briefly, the StageTip was prepared by packing three layers of reverse-phase SDB-RPS Empore™ disks in 1 mL pipette tips. Protein samples extracted in lysis buffer cocktail containing reducing and alkylating agents from cells were loaded into StageTip and digested for 16 h with Lys-C 1:100 (w:w, Lys-C:protein) and trypsin 1:50 (w:w, trypsin:protein). The digest sample is acidified with 0.5% TFA and then five volume of ethyl acetate was added and vortexed (500 × g, 2 min, room temperature). The resultant peptides were washed first with 1:1 ethyl acetate:0.2%TFA (v:v) and then by 0.2% TFA centrifuging (1000 × g, 1 min, room temperature). Peptides were then eluted by 80% ACN in 5% NH₄OH basic solution for single shot or in a stepwise varying % ACN for fractionation.

Methanol/chloroform protein precipitation was performed on the protein extracted from tissue samples[10]. For one volume of sample, four volumes of methanol, one volume of chloroform, and three volumes of ultra-pure water were sequentially added with intensive mixing. After centrifugation (10 min at room temperature, at 16,000 × g), the upper aqueous phase was removed without disturbing the interface and the precipitate was washed by three volumes of methanol with thorough vortexing. The sample was centrifuged (5 min at room temperature, at 16,000 × g) and the supernatant was removed. The white protein precipitate was then allowed to air dry. The extracted proteins were resuspended in 8 M Urea, then reduced by 10 mM DTT at 29 °C for 30 min, and alkylated with 50 mM IAM at 29 °C for 30 min in the dark with temperature selection aimed at reducing urea carbamylation[10,47]. The samples were diluted with onefold volume of 50 mM TEABC and Lys-C was added at a ratio of 1:100 (w:w, Lys-C:protein) for digestion at 29 °C for 3 h. The samples were further diluted with threefold volume of 50 mM TEABC for trypsin digestion with a ratio of 1:50 (w:w, trypsin:protein) for 18 h at 29 °C. The proteolytic digestion was quenched by adding 10% TFA to a final concentration of 0.5% in the sample. The resultant peptides were desalted by reversed-phase StageTips[22]. Briefly, SDB-XC (Styrene Divinylbenzene) membrane was packed into the 200 µL stage tip, activated with 80% ACN, and conditioned with 0.5% ACN. After the sample loading, washing was performed under 0.5% ACN. Finally, the peptides were eluted by 80% ACN and transferred into a new tube. The yeast protein extract was also digested in solution and desalted in StageTip.

**HpRP peptide fractionation.** Reversed-phase peptide fractionation was performed for tryptic peptides from cell lines and tissue using StageTip format[22] and HPLC column format[10], respectively. The StageTip was prepared by packing reversed-phase membranes styrene divinylbenzene resin modified with sulfonic acid group (SDB-RPS) membranes Empore™ disks into the Gilson 200 µL tips. For peptide fractionation from cell lysate, tryptic peptides obtained by in StageTip digestion were fractionated and eluted using buffers with increasing ACN percentage (10%, 15%, 20%, 30%, 45%, 60%, and 80%) prepared in 40 mM ammonium formate. StageTip was centrifuged at 1000 × g for 2 min for elution of peptides from each fraction. Eluted peptides were collected and dried in SpeedVac or sequentially followed Fe-IMAC for phosphopeptide enrichment.

For peptide fractionation from pooled tissue sample, peptide sample was re-dissolved in 0.6 mL 5 mM ammonium formate pH 10 and 2% ACN, and loaded on a 4.6 mm × 250 mm Zorbax 300 Å Extend-C18 column (Agilent, 3.5 µm bead size) at a flow rate of 0.5 mL/min on a Waters alliance e2695 HPLC instrument coupled with Waters 2489 UV/Vis detector and fraction collector III. Solvent A (2% ACN, 5 mM ammonium formate pH 10) and a nonlinear increasing concentration of solvent B (90% ACN, 5 mM ammonium formate pH 10) were used to separate peptides. A 120 min LC gradient run started with 100% solvent A for 10 min, then increased linearly in percentage of solvent B to 10% in 5 min, from 10% to 40% in 35 min, 40% to 60% in 30 min, and 60% to 90% in 12 min, with an 8 min hold at 90% solvent B. Peptides were collected every minute for a total of 96 fractions from 11 to 106 min and then divided into 90% and 10% for phosphopeptide enrichment and for proteome analysis, respectively. The fractions were then combined into 12 fractions for phosphoproteome and 24 or 32 fractions for proteome-level analysis with a stepwise concatenation, followed by desalting with SDB-XC StageTip and dried with SpeedVac and stored in −80 °C until phosphopeptide enrichment.

**Phosphopeptide enrichment by tip-based Fe-IMAC.** The phosphopeptide enrichment was performed by home-made immobilized metal affinity chromatography (IMAC) StageTip according to our previous reports with some modification[48,49]. The IMAC tip was capped at one end with a 20 µm poly-propylene frits disk (Agilent, Wilmington, DE, USA) enclosed in a tip-end fitting. The tip was packed with 20 mg of Ni-NTA silica resin (QIAGEN) dissolved in 6% acetic acid and loaded onto StageTip by centrifugation (3300 × g for 3 min at room temperature). All purification steps for buffer exchange and sample loading were performed by centrifugation. The Ni²⁺ ions were removed with 50 mM EDTA in 1 M NaCl. The tip was then activated with 100 mM FeCl₃ and equilibrated with loading buffer of 6% (v/v) acetic acid at pH 3.0 prior to sample loading. Tryptic peptides (typically 200 µg) were reconstituted in loading buffer and loaded onto the IMAC tip. After successive washes by 100 µL washing buffer (loading buffer: ACN = 3:1) and 0.5% acetic acid, the bound peptides were then eluted twice from IMAC tip with 100 µL of 200 mM NH₄H₂PO₄. Eluted peptides were desalted using reversed-phase StageTips, dried under vacuum, and reconstituted in 0.1% FA for LC-MS/MS analysis.

**Synthetic phosphopeptides analysis.** A total of 166 synthetic phosphopeptides selected from 61 protein groups associated with lung cancer-related driver genes were dissolved in water. The synthetic phosphopeptides were desalted in SDB-XC StageTip and pooled in equimolar concentration to prepare stock solution. Synthetic phosphopeptides were then mixed in five different amounts (2, 1, 0.5, 0.2, and 0.1 ng) and spiked into the 0.5 µg tryptic peptides from yeast lysate as background and analyzed in DIA mode in triplicates. Triplicate DDA files were also acquired to generate a spectra library. The DIA data were then processed against the spectral library of synthetic phosphopeptides and yeast peptide background using *S. cerevisiae* fasta (UniProtKB/Swiss-Prot database, February 2018 downloaded).

**LC-MS/MS analysis.** The LTQ Orbitrap Fusion Lumos Tribrid mass spectrometer (Thermo Fisher Scientific) coupled with an Ultimate 3000 RSLCnano system (Thermo Fisher Scientific) was used in this study. The mobile phases consisted of buffer A (0.1% FA in water) and buffer B (0.1% FA in ACN). Tryptic peptides were resuspended in 0.1% FA and spiked with iRT peptides according to the manufacturer's protocol. Peptides were then separated on Thermo Scientific PepMap C18 of 50 cm length, 75 µm inner diameter, packed with 2.0 µm particles of 100 Å pore size (Thermo Fisher Scientifc). The phosphopeptides were typically separated at 300 nL/min flow rate with 2% buffer B to 15 min, ramping from 5% to 28% to 118 min, 28% to 50% over 15 min, keeping at 95% from 135 to 140 min, decreasing to 2% from 142 to 145 min.

The MS instrument was operated in the positive ion mode with an electrospray through a heated ion transfer tube (250 °C). For DDA datasets, the spectra of full MS scan (400–1250 m/z) were acquired in the Orbitrap with MS resolution of 60,000 at m/z 200 Da for a maximum injection time of 50 ms with an automatic gain control (AGC) target value of 4e5. Fragment spectra were obtained in the higher-energy collisional dissociation (HCD) mode using a normalized collision energy of 30%, resolution at 15,000, injection time of 75 ms, and AGC target of 5e4. Top ten precursors were selected for MS2 analysis with an isolation window of 1.4 m/z and dynamic exclusion time set to 20 s. The MS DIA datasets were acquired using the following parameters: scan range = 400–1250 m/z, MS resolution of 60,000 at m/z 200, an AGC target = 4e5, and maximum injection time = 50 ms. The MS/MS scan was performed in HCD mode with the following parameters: using 10 Da isolation window with 1 Da overlap over 500–1000 m/z precursor and scan range of 110–1600 m/z, resolution 30,000 with maximum injection time of 54 ms, AGC target = 5.0e4, and normalized collision energy = 30%. All data were acquired in positive polarity and profile mode.

**Data processing and protein identification.** For DDA-based library, database search of all DDA data was performed by MaxQuant[50] (v1.5.3.30) sing Andromeda

search engine with parameters slightly modified from standard setting for Orbitrap instrument against the UniProtKB/Swiss-Prot database (2015_12 release, *Homo sapiens* = 20,193 entries) with inclusion of the 11 synthetic iRT peptides and β-casein standard protein sequences. Maximum of two missed cleavages were allowed for trypsin digestion with carbamidomethylation (+57.022 Da) of cysteine residues set as static modifications. Variable modification of phosphorylation (+79.966 Da) on Ser, Thr, and Tyr residues; oxidation of methionine (+15.995 Da) residues and acetylation on protein N terminus (+42.016 Da) were set. The tolerance for spectra search allowed 10 p.p.m. for precursor and 0.05 Da tolerance for fragment ions. Other parameters include modified score ≥ 40, delta score ≥ 8, intensity ≥ 100, and peptide length ≥ 7 amino acids. Protein and peptide were both filtered at global 1% FDR at PSM and protein levels, as well as phosphosite-level FDR. DDA LFQ was performed using MaxQuant with or without MBR features.

**Construction of hybrid phosphoproteome spectral library.** The hybrid phosphopeptide spectra library was generated by DDA of 156 raw files from fractionated cell line and cancer tissue samples, as well as 24 DIA raw files of fractionated cell lysates using Spectronaut Pulsar search (Biognosys, v14, Switzerland) according to Muntel et al.[23] report. All the DDA data were first searched in a default setting with variable modification as oxidation of methionine (+15.995 Da) residues and acetylation on protein N terminus (+42.016 Da), as well as phosphorylation (+79.966 Da) on Ser, Thr, and Tyr residues. Fixed modification of carbamidomethylation (+57.022 Da) of cysteine residues was included. A cutoff 1% FDR was set at PSM, peptide, and protein level activating phosphosite localization. Finally, a DIA data (*n* = 24) raw file was processed similarly combining with the PSM search archive of the DDA files to generate a library with a combined FDR estimation. Fragment ions minimum *m/z* 300, maximum *m/z* 1800, minimal relative intensity of 5%, and 15 most intense fragment ions per precursor were included and those with less than three amino acid residues were not considered. Fragment ions with neutral losses were included. Normalized retention time was obtained using segmented regression to determine iRT in each run by the precision iRT function. Precursors with phosphorylation modification as well as non-modified were finally retained.

**Sample preparation and data processing for proteome spectra library construction.** An independent proteome spectral library was constructed from 22 lung cancer tumor tissues pooled in 2 batches and 5 NSCLC cell lines analyzed individually. For each cell lysate and the two pooled tissue batches, the methanol/chloroform precipitated proteins were digested as described above and peptides from each cell lysate were fractionated by HpRP in StageTip using C18 bead, whereas peptides from pooled tissue samples were fractionated by HPLC column as described above. The iRT peptides (Biognosys) were spiked in each sample followed by LC-MS/MS analysis using LTQ Orbitrap Fusion Lumos Tribrid mass spectrometer (Thermo Fisher Scientific) coupled with an Ultimate 3000 RSLCnano system (Thermo Fisher Scientific). Peptides were separated on Thermo Scientific PepMap C18 column of 50 cm length, 75 μm inner diameter packed with 2.0 μm particles of 100 Å pore size (Thermo Fisher Scientific). Peptides were separated at 300 nL/min flow rate with 2% buffer B to 20 min, ramping from 5% to 31%, to 120 min, 31% to 45% over 12 min, keeping at 95% from 135 to 140 min, decreasing to 2% from 142 to 145 min was used. For DDA data, the spectra of full MS scan (375–1600 *m/z*) were acquired in the Orbitrap with MS resolution of 60,000 at *m/z* 200 for a maximum injection time of 50 ms with an AGC target value of 4e5. Fragment ion spectra were obtained in the HCD mode using a normalized collision energy of 30% with resolution at 15,000, injection time of 50 ms with AGC target of 5e4. Top 15 precursors were selected for MS2 analysis with an isolation window of 1.4 *m/z* and dynamic exclusion time was set to 20 s. A total of DDA 191 raw LC-MS/MS datasets were generated. The data were first processed by MaxQuant for protein identification at 1% PSM and protein FDR, and then imported to Spectronaut to construct a library. The standard parameters in Spectronaut was used with a maximum of six most intense fragments per precursor (minimum six) included.

The single-shot DIA dataset of proteome profiling from cell line and tissue samples proteome-level quantification were similarly acquired over the same instrument platform. The peptides were separated at 400 nL/min flow rate with 2% buffer B to 20 min, ramping from 5% to 31%, to 175 min, 31% to 45% over 20 min, keeping at 95% from 210 to 220 min, decreasing to 2% from 220 to 222 min. The MS DIA data was acquired with the following parameters: scan range = 400–1250 *m/z*, MS resolution of 120,000 at *m/z* 200, an AGC target = 4e5, and maximum injection time = 50 ms. The DIA-MS/MS scan was performed in the HCD mode with the following parameters: isolation window of 10 Da with 1 Da overlap, precursor range = 400–1000 *m/z*, fragment scan range 110–1600 *m/z*; resolution = 30,000 with maximum injection time of 54 ms, AGC target = 5e4; normalized collision energy = 30%. All data were acquired in profile mode using positive polarity.

**DIA data analysis.** DIA data signal extraction and quantification were performed with analysis pipeline in Spectronaut™ (Biognosys, v14)[24] using standard setting with some modifications. In brief, dynamic retention time prediction with local regression calibration was selected. Interference correction on MS and MS2 level

was enabled. The FDR was set to 1% at peptide precursor and protein level using scrambled decoy generation and dynamic size at 0.1 fraction of library size. MS2-based quantification was used, enabling local cross-run normalization. Phosphopeptide precursor abundances were measured by the sum of fragment ion peak areas and peptide grouping was performed based on modified peptides. Phosphosite localization tool recently integrated into Spectronaut was applied to filter class 1 localization and a class 1 probability cutoff ≥ 0.75 for DDA and libDIA, and ≥0.99 for dirDIA employed[19]. For site-specific quantification, the abundance of a phosphorylation site was calculated by summing up all the abundances in precursors/peptides, which contain the site in the corrected precursor/peptide abundance file. Protein-level quantification was performed against the proteome library in a standard setting with stripped peptide sequence area obtained from the mean precursor area.

**Statistical analysis and pathway annotation.** Further statistical analysis was performed by Perseus software (1.6.1.1)[51] All the phosphosite abundance was performed on the logarithmic (log₂) ratios. FDR controlled two-sample *t*-test was performed for NSCLC EGFR-TKI-sensitive and -resistant cell lines (permutation-based FDR < 0.01 and S0 = 0.1). Statistical significance of changes in abundance between tumor and normal groups was calculated by paired two-sample *t*-tests (*p* < 0.05, S0 = 0.1). Pathway-enrichment analyses was performed by KEGG[25]. Protein–protein interaction network and functional annotation was done by STRING[52] database (version 11). List of deposited phosphosite and kinase–substrate site pairs were obtained from PhosphositePlus[28], whereas linear kinase motif was enriched by generating 13 amino acid sequences with a phosphorylated residue at the center. To visualize the amino acid composition of the identified phosphopeptides, the motif logo was generated using pLOGO (v1.2.0)[53] and phosphorylation motif enrichment was performed based on Fisher's exact test (FDR < 0.02). Site-specific quantification was performed by in-house customized R scripts.

**Reporting summary.** Further information on research design is available in the Nature Research Reporting Summary linked to this article.

## Data availability
The mass spectrometry raw datasets, reference spectral libraries, Spectronaut quantification reports, and MaxQuant search results were deposited in Japan ProteOme Standard Repository[54] jPOST and can be accessed through ProteomeXchange[55]. For phosphoproteome dataset the accession number is P7 in ProteomeXchange and J9 in jPOST, respectively. For proteome dataset the accession number is P6 in ProteomeXchange and J4 in jPOST. Descriptions of the raw files is provided in Supplementary Data 8 file. Data used for Fig. 5g is available as Supplementary Data 6, whereas data for Fig. 6a, c, f, g are available as Supplementary Data 7. The protein sequence fasta file was obtained from UniProtKB/Swiss-Prot database. The iRT peptides fasta file was downloaded from Biognosys website. The functional and family annotation was analyzed in STRING database (version 11), KEGG database, PhosphoSitePlus database, human dephosphorylation database DEPOD, and kinase families in KinMap database. Source data are provided with this paper.

## Code availability
Custom R code used for site quantification based on site annotation provided by Spectronaut was deposited in the jPOST repository. It can be accessed in ProteomeXchange through P7 and in jPOST through J9.

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

## Acknowledgements

This work was supported by the Ministry of Science and Technology, Taiwan (MOST 107-2113-M-001-023-MY3 to Y.J.C.) and Academia Sinica, Taiwan (AS-TP-108-M06 to Y.J.C.). This work was funded in part by US National Institute of Health grants (R01-GM-094231 and U24-CA210967 to A.I.N.). Tumor and adjacent normal tissue of lung cancer patients were supported by the Taiwan Cancer Moonshot Project in the Next-generation Pathway of Taiwan Cancer Precision Medicine Program (AS-KPQ-107-TCPMP) at Academia Sinica, Taiwan.

## Author contributions

R.B.K., Y.J.C., and P.Y.L. designed the research. R.B.K., P.Y.L., Y.C.C., and B.S.C. participated in the data generation. R.B.K., W.C.K., C.F.T., B.S.C., T.Y.S., and A.I.N. participated in data analysis. Y.J.C. and R.B.K. wrote the manuscript with inputs from all authors.

## Competing interests

The authors declare no competing interests.
