## [Peer Review File · Nature Communications]

REVIEWER COMMENTS

Reviewer #1 (Remarks to the Author):

The authors of the manuscript entitled “A Data-independent Acquisition-based Global Phosphoproteomics System Enables Deep Profiling” reported a global phosphoproteomics system (GPS) strategy by integrated data-independent acquisition (DIA) mass spectrometry and phosphosite localization for accurate site-specific quantification. After constructing a hybrid spectral library based on both library-based libDIA and library-free dirDIA data, the authors further evaluated the quality of their database using 160 synthetic phosphopeptides. They next applied their system to profile the phosphoproteome in drug-resistant cells and patient-derived cancer tissues.

In general, the main significance of this study is the construction of the biggest phosphopeptide spectral library of lung cancer up to date, which achieves a high phosphoproteome coverage for DIA MS analysis. Conceptually, this approach could be valuable to facilitate the characterization of phosphorylation signaling events in cancers due to the increased detection sensitivity and quantification reproducibility. Nevertheless, the results in many parts of the manuscript do not clearly demonstrate its methodology advancement or potentiality for the discovery of new biological function/mechanism. Some key experiments could not convincingly support their conclusion, which need to be clarified and redesigned.

Major concerns:

1. In page 6, the rationale to calculate the phosphosite localization probability from DIA data is unclear. The authors gave an example of the calculation process in the Supplementary Note. It was not clear what the general principle of such calculation was? There is no convincing result to support the high accuracy of site localization and quantification in such way.
2. In page 7, the receptor tyrosine kinase (RTK)/RAS/RAF and its downstream kinases are frequently mutated in lung cancer, and currently the most successful approach for targeted therapy. What is the rationale that the authors chose the 160 synthetic phosphopeptides? Do they cover all the primary components and druggable targets in the signaling pathways of lung cancer?
3. Also in this section, one example of mono- and multiple phosphopeptides of the “GSHQISLDNPDYQQDFFPK” sequence from EGFR alone (Fig. 2a, 2b) is not convincing enough to demonstrate the overall DIA spectral quality for unambiguous site localization.
4. In page 8, for the coverage of the phosphotyrosine-enhanced hybrid library (Fig.3d-f), how many class 1 sites were identified in these pathways? Probability score ≥ 0.75 is a general criterion to dissect the biological function of a phosphorylation site from a phosphoproteome data. It remains unclear how many key phosphosites were identified in the RTK/RAS/RAF pathways and other targetable proteins. This is critical and helpful to understand lung cancer mechanism and identify therapeutic targets.

5. In page 8, The authors found only 22.6% phosphopeptide overlap between cell lines and NSCLC tumor samples. It was somewhat surprising. Why the overlap is so low between the samples with tissue origin?

6. In page 11, the authors constructed the database by using the samples only originating from NSCLC (cell lines and patients), but they further evaluated its performance in breast cancer. I do not see the value of this experiment. As I mentioned in the above comment, even the phosphopeptide overlap between samples from the same tissue origin is so low. The tissue-specific phosphosite loss could be huge. Can the authors justify the value of this strategy for phosphoproteome analysis of other types of cancer?

7. The main conclusion of “single-shot DIA offers highly reproducible large-scale phosphosite quantification” section is that libDIA could achieve higher phosphosite identification and higher quantification reproducibility than DDA method. These conclusions are well-known merits of DIA method itself. What is the advantage/disadvantage of GPS strategy over other DIA methods? This question should be clearly addressed in this study.

8. In page 13, the section of phosphoproteomics profiling of EGFR-TKI-sensitive and resistant lung cancer cells is largely descriptive. The result of this section does not provide a clear methodology advancement or new mechanistic insight in this type of study. The experimental design remains questionable, either. There are many different mechanisms for intrinsic/acquired resistance for EGFR inhibitors, such as EGFR target alterations, bypass tracks (MET amplification, HER2 amplification, NF1 loss, etc.). Why the authors chose to compare two cell lines with different genetic backgrounds? What is the known resistant mechanism for CL68 cell? What is the new information that GPS strategy could provide? Does the high phosphotyrosine proteome coverage of this strategy contribute to the new mechanistic understanding of TKI resistance?

9. In page 15, the questions remain similar for phosphoproteome profiling in lung cancer tissues. The result is largely descriptive. The experimental design remains unclear. The authors chose 3 early-stage tumors and 2 late-stage tumors. Lung cancer is of high heterogeneity. There are so many factors that impact lung cancer pathogenesis, such as cancer driver genes, age, gender, smoking status, environment, etc. Such a small number of samples could be very difficult to yield biological insight. Again, what is the new information or advancement that GPS strategy could provide for clinical sample analysis? The authors published a large-scale TMT-based proteogenomics study of lung adenocarcinoma clinical samples (Cell. 2020,182:226-244). What is the advantage of GPS strategy over the TMT-based approach for clinical sample analysis? A detailed comparison between these two studies is necessary.

Minor concerns:

1. The authors collected 25 tumor samples, but only 22 of them were used for spectral library construction. The selection criterion need to be provided.

2. The authors cultured five cell lines in the Methods section but only four were used for database construction. The selection criterion need to be provided.

3. In Fig 2A, b9-98 peak could not be found on the MS/MS spectra of the modified peptides with phosphorylation in Y1172.

4. What is the criterion to define up-regulated or down-regulated phosphosites in the last two sections?

5. The authors collected cells and heated them at 95 °C for 5 minutes. Did the author consider effect of high temperature on phosphorylation status? In addition, the authors chose 29 °C for protein alkylation and trypsin digestion, which was an unusual reaction condition. The authors have to justify such procedure.

6. The authors used both phosphosite plus and UniProtKB for the analysis of their data, but the criterion for these two databases was not clearly presented.

Reviewer #2 (Remarks to the Author):

The authors Kitata et al describe the development of a large lung cancer-specific phosphopeptide library and apply it to measure phosphorylation signaling in lung cancer cell lines using DIA. In general, the manuscript is well written and well reasoned. I have some concerns about how localization was performed with PhosphoRS, given that the tool was originally designed for DDA.

In particular, I am somewhat concerned that fragment ion reports generated by most DIA software do not include co-eluting fragments (and noise) that were not assigned to the "detected" peptide. These fragments and noise are crucial for the PhosphoRS localization algorithm to perform correctly, since it is based on the binomial probability of randomly finding fragment ions given the background ion frequency. In particular, co-eluting alternate localization forms will generate localization-important ions that are not considered in these reports. The absence of these ions will provide a false sense of security that the "detected" peptide has no other competing alternate localization possibilities. As a result, the localization probabilities generated by PhosphoRS may appear accurate in the control experiment with only a few, known phosphopeptides in a background of unphosphorylated yeast peptides, but not actually be accurate in the presence of a complex phosphopeptide matrix with more co-eluting phosphopeptides.

In addition, the authors do not discuss quantification using only site localizing ions, which may complicate the measurements of co-eluting phosphopeptide positional isomers. Both of these concerns would be somewhat mitigated if the authors demonstrated the fraction of localized phosphopeptides that have potentially overlapping elution patterns using their methods.

Finally, I have the following minor comments:

1) Page 4, paragraph 1: This section on phosphoproteomics with DIA should cite Rosenberger et al 2017 (PMID: 28604659) and Meyer et al 2017 (PMID: 28661500) as previous efforts to analyze and site localize phosphopeptides. In addition, this section should also cite Lawrence et al 2016 (PMID: 27018578), which demonstrated the construction of a similar large-scale phosphopeptide library with applications in DIA.

2) Page 6, line 128: The methods section discusses library creation in more detail, but it would be beneficial to mention here that the MaxQuant DDA library was generated with Global FDR filtering. Interestingly, the methods section does not discuss site-level FDR but it is mentioned briefly here. The methods should reflect this filtering as well.

3) Page 6, line 133: The tool used for DIA analysis (Spectronaut) should be mentioned here, because this citation (16) suggests that DIA-Umpire was used.

4) Page 6, line 139: More details should be added about how PhosphoRS was adapted for DIA.

5) Page 8, line 170: Should "predicted retention times" be "library retention times"? The "predicted" language may confuse people into believing that the retention times were derived de novo (e.g. with SSRCalc or Prosit).

6) Page 8, line 186: This should indicate which cell lines were used. Were they the same as PC9 and CL68 from later in that paragraph?

7) Page 11, line 236: Was PhosphoRS used to localize peptides for both DIA and DDA, or was the built in MaxQuant localization algorithm used? If so, are these comparable since they produce very different types of scores?

8) Page 11, line 249: This section indicates that both direct and library searches were combined to identify phosphorites. Were they FDR corrected together afterwards, or is the actual FDR inflated from errors in both searches? How are the search and localization scores comparable since they were derived using different types of data (e.g. the DIA reports may have no alternate isomer localization ions [see above] or noise, while the DDA reports have both noise and alternate ions)?

9) Page 14, line 319: The motif enrichment tool that was used should be indicated and cited.

10) Page 23, paragraphs 2 and 3: Two types of peptide fractionation were performed: stagetip fractionation and fractionation with an HPLC. The samples that were prepared with each type of fractionation should be indicated. Additionally, these sections should be more clearly delineated so that it's clear these are two types approaches.

11) Page 24, line 549: How were the synthetic peptides made? Where they purchased? From whom?

12) Page 26, line 580: The MS/MS maximum injection time setting should be indicated for the DIA method.

13) Supplementary Note, page 6: The discussion of the PhosphoRS reimplementation indicates the use of phospho neutral losses (e.g. -98 Da) for localization. These ions are generally considered untrustworthy for localization because they can also be generated from unphosphorylated series and theonines through commonly occurring loss of water (-80,-18).

Reviewer #3 (Remarks to the Author):

== Conclusion ==

In their manuscript submission, Kitata et al. describe a comprehensive MS DIA strategy to both acquire and process deep profiling phospho-proteomics data. They combine a well-established and robust phospho-proteomics workflow with novel approaches in data analysis (hybrid DDA + DIA spectral library; hybrid dirDIA + libDIA analysis; phosphoRS phospho-site-localization for DIA data) to generate the deepest phospho-proteomics spectral library for a single cancer type. This is an impressive and extensive amount of work.

From my perspective, the biggest strength of the manuscript is its deep, high quality phospho-proteomics spectral library, acquired by samples generated and measured by the same group on a single type of MS instrument. High quality phospho-proteomics spectral libraries cannot yet reliably be generated in silico such as proteome spectral libraries, and as such are rare and have many uses in the community. Furthermore, it seems to be the deepest generated for a single human cancer type (NSCLC). At the same time, this may also limit its most comprehensive coverage to NSCLC samples, as phospho-peptides from other cancers / tissues may not be represented. For these types of samples, a broad coverage library such as the “plug-and-play” library generated by Lawrence et al. (2016), which covers 109,611 phosphorylation sites from a variety of cell types, might be more applicable.

On the data analysis side, the adaptation of the publicly available phosphoRS localization tool to DIA input data further provides a good alternative to the already available DIA PTM localization solutions Thesaurus (Searle 2019) and Spectronaut (Bekker-Jensen 2020). The novel ways to generate DDA/DIA hybrid spectral libraries and merge the output of Spectronaut dirDIA and libDIA I find intriguing, but have a few concerns pointed out in the comments below.

All in all, I believe this study and the high-quality data presented in it will be of great value to the phospho-proteomics community. For this reason, I would recommend publication in Nature Communications with revision of the points below.

== Major Comments ==

1. The authors state in their study that the generation of a hybrid spectral library (DDA + DIA runs) may provide gains in peptide identifications (IDs) compared to a single type library. Even though I think this is an intriguing hypothesis, no data is provided to compare advantages/disadvantages of a hybrid vs a single DDA or single DIA data. However, could the mix of DDA and DIA fragmentation patterns even be problematic, as it provides two different types of data during machine learning identification of peptides in DIA runs? Would it not be better to see if DDA or DIA runs are better for library generation, and the measure all library runs the same way?

2. Connected to point 1, lines 613-614 do not indicate FDR re-processing after merging of DDA and DIA into the hybrid library. If so, could this inflate the library with false positive IDs?

3. The authors present both in Fig 4 and the abstract, that they merge the IDs of the dirDIA and libDIA analyses to get even higher ID analysis results. Similarly, as in point 1, could this inflate the number of false positive IDs, since results from different analysis pipelines were not subjected to a

common FDR? In the worst case, could the gain of IDs over the individual dirDIA/libDIA approach be attributed to false positive IDs? (If so, the approach might still be useful to generate extremely robust IDs, by filtering for the overlap only - similar as in PeptideShaker by Vaudel et al. 2015.)

4. The authors claim that they generated the deepest DIA spectral library for a single cancer tissue type (NSCLC). Since the plug-and-play library generated by Lawrence et al. (2016) covers substantially more phospho-peptides, could an analysis using this library reach the same depth and performance on the NSCLC samples?

5. It is not always clear for which statistical tests and figures FDR control was applied. In general, statistical tests on proteomics data should always include multiple testing correction. Lines 684-688 read as if FDR control was only used for TKI test, but not for tumor-vs-normal test. Fig 5f shows $-\log(p\text{-value})$, which indicates no multiple testing correction was applied. Was FDR control not applied in these cases? If so, why not?

6. The authors do not present a discussion of limitations in their discussion section. If they agree that some of the points raised here are valid, they might want to consider discussing them.

== Minor Comments ==

7. The authors use pervanadate to increase phospho-tyrosine IDs. Could it be that due to chemical induction, the gain in IDs mostly reflects phospho-peptides not relevant in biological settings? Would an enrichment using anti-phospho-tyrosine antibodies be more appropriate?

8. Since the data was already analysed using Spectronaut, would it be possible to achieve the same localization performance presented by DIA phosphoRS using the Spectronaut localization approach (Bekker-Jensen et al. 2020)?

9. For the MaxQuant phospho-peptide DDA analysis, match-between-runs (MBR) was used. How was localization filtering performed here, since MBR does not provide localization information (Fig 4a)?

10. It seems that some samples were used both for library construction AND as samples themselves (e.g. tumor samples 21 and 24, supp table 1b). Were these measured as independent replicates, or could they otherwise lead to overfitting of data?

11. It could be beneficial to point out that Fig 5 phosphorylation intensities are normalized on protein levels in the figure legends, since this directly influences how to interpret the data. Importantly, it is not clear in Fig 6 if this data was also protein-level normalized or not.

12. The authors might want to consider exchanging their pie chart plots (Fig 3c, 3d) with other appropriate charts (e.g. bar), as pie chart data is difficult to interpret/compare.

13. Based on references in the text (lines 260 and 272), it seems that supplemental figures 4 and 5 are displayed in the wrong order.

14. Data import into Excel seems to have caused date conversion of gene names (e.g. Supplemental table 5a line 16599: gene "SEPT6" was renamed "6-Sept") -> to facilitate unproblematic reanalysis of data, consider reimporting/correcting these cases (see Ziemann, Eren and El-Osta 2016)

15. The authors provide an extensive list of samples used, and a detailed materials and methods part. However, it would be great to also have a table listing which raw files were used for which analysis (libraries and samples) / which figure, as the raw files have arbitrary names. Additionally, it would be great to have the custom R scripts and Perseus session files published on public repositories to allow others to understand the statistical tests applied.

16. If R packages were used that require citation, these should be cited in the material and methods section.

A Data-independent Acquisition-based Global Phosphoproteomics System Enables Deep Profiling

We thank all reviewers for their kind and constructive comments and suggestions to improve our manuscript. We have attempted to address the comments and questions either in the body of the revised manuscript, or in the point-by-point reply letter. We hope that our revisions and explanations adequately respond to the comments and clarify and enhance the manuscript. The point-by-point responses to Reviewers' comments are listed in a separate "point-by-point response" file. For your information, the additional analysis and major revisions are summarized as follows:

1. Reconstruction of hybrid phosphoproteome library

1.1 To provide the common FDR control for confident phosphopeptide identification, we have reconstructed a hybrid phosphoproteome library of combined datasets (n=180 raw files) at 1% FDR cutoff (PSM, peptide & protein) by using recently updated Spectronaut Pulsar (v14). This resulted in a spectra library of 159,524 phosphopeptides (121,407 class 1) corresponding to 88,107 phosphosites on 8, 805 protein groups. Furthermore, we provide the information on highly confident class 1 phosphosites in the revised manuscript.

1.2 We appreciate that the Reviewer pointed out the novelty of our hybrid library. To further demonstrate its advantage, we have added new analysis to show that superior performance improvement can be achieved with 12-40% more phosphopeptides by using a hybrid phosphoproteome library compared to a library constructed from single data type (DDA or DIA).

2. Update of lung cancer proteome library and DIA analysis

2.1 By using the newly launched Spectronaut V14, we also reconstructed the proteome spectral library (n=191 DDA raw files) from lung cancer cell lines and tissues. The library consists of 223,091 peptide sequences of 12,344 protein groups, which represents the deepest proteome library from single sample type over single MS platform.

2.2 We added data of performance evaluation of proteome library to show strength of single shot DIA compared with DDA using lung cancer cell lines.

3. Comparison with large-scale phosphoproteome database

We have added comparison of our GPS library with large-scale phosphoproteome dataset and databases including HeLa phosphoproteome (Sharma et. al., *Cell Rep.* 2017), Plug-and-play library for phosphoproteome (Lawrence et.al. *Nat. Methods* 2016) and PhosphositePlus database. Compared to the large scale datasets of HeLa phosphoproteome and Plug-and-play library, our

GPS library provided 40,559 (28%) additional phosphosites and 85,716 unique phosphopeptides, respectively. Compared to the largest phosphosites database, PhosphositePlus, additional 26,234 sites (17,097 class 1) were newly identified by our GPS.

4. **Incorporation of phosphosite localization tool from Spectronaut**

To ensure the phosphosite confidence in our library and analysis, we have adapted the new DIA-specific PTM site localization function in Spectronaut V14 (Bekker-Jensen et al. *Nat. Commun.*, 2020) to compute site confidence. All the datasets were reanalyzed; Figures and Results were extensively revised accordingly.

5. **Validation of site localization and quantitative performance using 166 Synthetic phosphopeptides**

We also reconstructed synthetic phosphopeptide library (166 phosphopeptides) and re-analyzed the site localization using Spectronaut (V14). Phosphosite localization analysis indicated that DIA spectra of 161 phosphosites (96%) have a high accuracy for determining the confident class 1 sites (probability ≥ 0.75). With a very high stringency of 0.99 probability cut-off, 142(90.4%), 135(86%), 122(77.7%) and 118(75.2%) were quantified from 2, 1, 0.5, 0.2 and 0.1 ng of pooled amount. We further showed the potential of DIA for the confident identification, site localization and quantification of multiple phosphorylated peptides.

6. In our previous manuscript, we use the non-small cell lung cancer pathway, which is the key pathway for Asian patients, to illustrate our method performance. To provide broader biological insight from our findings, we added a new figure to show highly confident class 1 phosphosites coverage in the **RTK-RAS/RAF pathway** that is globally important for lung cancer.

7. To address reviewers' comments, we also provided comparison between our DIA dataset and TMT results as well as comparison of Spectronaut based and PhosphoRS site localization. In summary, compared to the TMT-based strategy, our single-shot DIA analysis demonstrated strengths in high sensitivity, deep coverage, high efficiency, low missing values as well as resolving the ratio compression problem.

Following the above experiments, **we have extensively revised ALL the Figures (Fig. 1-6; Supplementary Fig. 1-9) and Supplementary Table 2-8) and manuscript accordingly**. The point-by-point responses to all three Reviewers' comments are listed below in **blue font** as well as in a separate "point-by-point response" file. The major revision and updates are indicated in the revised manuscript with **blue color font**.

Reviewer #1 (Remarks to the Author):

The authors of the manuscript entitled “A Data-independent Acquisition-based Global Phosphoproteomics System Enables Deep Profiling” reported a global phosphoproteomics system (GPS) strategy by integrated data-independent acquisition (DIA) mass spectrometry and phosphosite localization for accurate site-specific quantification. After constructing a hybrid spectral library based on both library-based libDIA and library-free dirDIA data, the authors further evaluated the quality of their database using 160 synthetic phosphopeptides. They next applied their system to profile the phosphoproteome in drug-resistant cells and patient-derived cancer tissues.

In general, the main significance of this study is the construction of the biggest phosphopeptide spectral library of lung cancer up to date, which achieves a high phosphoproteome coverage for DIA MS analysis. Conceptually, this approach could be valuable to facilitate the characterization of phosphorylation signaling events in cancers due to the increased detection sensitivity and quantification reproducibility. Nevertheless, the results in many parts of the manuscript do not clearly demonstrate its methodology advancement or potentiality for the discovery of new biological function/mechanism. Some key experiments could not convincingly support their conclusion, which need to be clarified and redesigned.

Response: We thank the reviewer for the comments to our manuscript and recognizing the strength of increased detection sensitivity and quantification reproducibility using our constructed phosphoproteome resource library. After constructing the deep phosphoproteome and proteome libraries, our major efforts focus on demonstrating the applicability from a 166 synthetic phosphopeptides, cell lines to more complex patient derived tissue samples. For its utility for clinical proteomics research, we think the high coverage of cancer-related pathways and enhanced tyrosine phosphosites will be crucial. For example we were able to identify 12 tyrosine sites of EGFR, including the key autophosphorylation sites of Y1197 and Y1172, without the use of antibody. We believe that the resource and strategies provided can be useful tools to study cancer biology.

To address the major comments, we performed the following major revisions:

(1) We have reconstructed the hybrid phosphoproteome library with a combined FDR cutoff [Muntel et al., *Mol. Omics*, 2019] and demonstrated enhanced coverage by using hybrid library compared to spectra libraries established from single acquisition mode (DDA or DIA). (**Fig. 3**, and **Supporting Fig. 3; Supplementary Table 3**)

(2) We incorporated phosphosite localization probability calculation by the updated version of Spectronaut (Bekker-Jensen et al., *Nat. Commun.*, 2020) and re-analyzed all the data in this study. Using the new Spectronaut (V14) search for phosphoproteome (n=180 raw files), 121,407 (76%) among 159,524 phosphopeptides have class 1 localized sites (probability of 0.75 or higher) in our hybrid library (**Fig. 3b; Supplementary Table 3**).

(3) To provide insight on the RTK-RAS/RAF pathway, we added **Supplementary Fig. 4** to show the coverage of high confident class 1 phosphosites.

(4) We added the result on identification of class 1 phosphosite (161 phosphosites, 96%) among 166 synthetic phosphopeptides. The result showed the potential of DIA for confident identification and quantification of more challenging multiple phosphorylated peptides (**Supplementary Fig. 2**).

Major concerns:

1. In page 6, the rationale to calculate the phosphosite localization probability from DIA data is unclear. The authors gave an example of the calculation process in the Supplementary Note. It was not clear what the general principle of such calculation was? There is no convincing result to support the high accuracy of site localization and quantification in such way.

Response:

1.1 When the manuscript was prepared (Spectronaut version 12), the available DIA tools provide phosphoproteomic identification without phosphosite localization. In this study, we demonstrated that a library-based approach achieved deep phosphoproteome profiling. To enhance the site localization confidence, in the previously submitted version of manuscript, we adapted site localization tools using the rich fragments peak lists generated by DIA method in accordance with the PhosphoRS which was commonly used for DDA dataset (Taus et al. *J. Proteome Res.* 2011). Following the approach of PhosphoRS, the fragment features including mass accuracy and peak area obtained from DIA spectra were used to calculate the site localization probability to distinguish phosphopeptide isomers. Compared to the commonly used 6 fragments per precursor, we used top 15 fragments per precursor to ensure better accuracy in the previous manuscript.

1.2 The new version of **Spectronaut** (version 13, 14) was recently launched and added the site localization function (Bekker-Jensen et al., *Nat. Commun.*, 2020). In the revised manuscript, we have adapted Spectronaut to compute the site localization confidence for all the dataset, library reconstruction and DIA quantification. Using the same software version and data from two cell lines of PC9 and CL68, comparable identification of class 1 phosphopeptide and sites could be achieved with 72% and 75% overlap, respectively from our previous strategy of PhosphoRS and Spectronaut based site localization (Figure below).

Figure. Comparison of site localization by PhosphoRS and Spectronaut using PC9 and CL68 cell. (a) Class 1 localized phosphopeptide precursors. (b) Class 1 localized phosphosites from the two cell lines. Triplicate DIA files of each cell line were analyzed using Spectronaut (v14) for signal extraction against GPS library (originally reported in the manuscript). Then site localization was performed using Spectronaut and by PhosphoRS. The result indicated 72% and 75% overlap in precursor and phosphosites, respectively.

1.3 The phosphosite localization confidence and quantification accuracy were evaluated by 166 synthetic phosphopeptides of known mono ($n=139$) and di ($n=21$) and tri ($n=6$) phosphorylation sites including several competing sites. These selected phosphopeptides were originally confirmed to be identified by DDA mode for library construction. The 166 phosphopeptides were pooled in 5 amounts (2, 1, 0.5, 0.2 and 0.1 ng) and spiked into 0.5 μg yeast tryptic peptides. The phosphosite localization analysis indicated that **96% DIA spectra of phosphosites (161) have a high accuracy for determining the class 1 sites** (probability ≥ 0.75). Overall, among 157 phosphopeptides within the DIA scanning m/z range, 153 and 143 phosphopeptides could be identified at localization probability of 0.75 and 0.99, respectively (**Fig. 2d**). The quantitation results show median ratios of 0.95, 1.8, 3.4, 8.7, and 20.1 for the expected ratios 1-, 2-, 4-, 10- and 20-fold, respectively, with CV% of 2.2-6.6%, demonstrating high reproducibility. The manuscript was revised accordingly.

1.4 To further show the confidence of accurate site localization, we have added the analysis results for multiple competing sites of di-phosphorylated peptides and tri-phosphorylated peptides in the **Supplementary Fig. 2** (shown below).

The following descriptions were also added to the manuscript

Page 8, line 183-188, "The localization probability of 222 precursors detected in dilution series and *the phosphosite localization result on the di- and tri-phosphorylated peptides with multiple competing sites at the same peptide sequence were shown in Supplementary Fig. 2. Among di-phosphorylated peptides, 18 out of 21 were confidently identified and quantified as class 1 with even in diluted concentration, and 4 of the 6 tri-phosphorylated peptides could be identified and quantified. (Supplementary Fig. 2). These examples demonstrate good quality DIA spectra for unambiguous site localization*".

Supplementary Figure 2

Supplementary Figure 2. DIA performance evaluation for multiple-phosphorylated peptides. (a) Overall phosphosite localization across dilution series of quantified 222 precursors with average of the triplicate localization data shown. Site-specific localization of (b) diphosphorylated; (c) triphosphorylated peptides.

2. In page 7, the receptor tyrosine kinase (RTK)/RAS/RAF and its downstream kinases are frequently mutated in lung cancer, and currently the most successful approach for targeted therapy. What is the rationale that the authors chose the 160 synthetic phosphopeptides? Do they cover all the primary components and druggable targets in the signaling pathways of lung cancer?

Response:

2.1 In East Asia, EGFR activating mutations (two major types: L858R point mutation and the E746_A750 exon 19 deletion) occur much more frequently (>60%, especially in never-smoker females) and patients bearing EGFR mutations benefit from targeted therapies using tyrosine kinase inhibitors, although most of them eventually develop resistance (Yang et al., *Annu. Rev. Med.*, 2020; Shi et al., *J. Thorac. Oncol.*, 2014; Chen et al., *Cell* 2020). **EGFR signaling cascades represent the most important pathways for lung cancer in East Asia.** Thus, we selected synthetic phosphopeptides from phosphoproteins in the lung cancer signaling related pathways, including *NSCLC signaling* (61 phosphopeptides, 58 sites), *EGFR-TKI resistance* (85 phosphopeptides, 80 phosphosites), *MTOR signaling* (37 phosphopeptides, 33 sites) and *PI3K-AKT signaling* (73 phosphopeptides, 69 sites).

2.2 For **RTK/RAS/RAF pathway**, in addition to 19 phosphopeptides of EGFR, we also selected GRB2 BRAF, MAPK1, and MAPK3. Other known druggable targets, including MTOR, MET, SRC and EML4, were also selected to obtain synthetic phosphopeptides. Accordingly, the following description was added to the text.

Page 7, line 162-170 *“The EGFR-initiated signaling cascades represent the most important pathways for lung cancer in East Asia. Thus, the synthetic phosphopeptides were selected from proteins in the lung cancer related signaling pathways, including NSCLC signaling (61 phosphopeptides, 58 sites), EGFR-TKI resistance (85 phosphopeptides, 80 phosphosites), MTOR signaling (37 phosphopeptides, 33 sites) and PI3K-AKT signaling (73 phosphopeptides, 69 sites). In addition to 19 phosphopeptides of EGFR, other RTK and drug targets including SRC, GRB2, BRAF, MTOR, MAPK1, MAPK3, MET and EML4 were also selected to obtain synthetic phosphopeptides (Supplementary Table 2).”*

2.3. For evaluation of technical performance, 27 phosphopeptides of multiple sites on the same sequences were designed to evaluate site localization capability. In summary, a total of 157 phosphopeptides were in the scanning m/z range (167 sites) of the DIA-MS method and they were all detected. Please see **Supplementary Table 2** for the list of phosphopeptide sequences.

3. Also in this section, one example of mono- and multiple phosphopeptides of the “GSHQISLDNPDYQQDFFPK” sequence from EGFR alone (Fig. 2a, 2b) is not convincing enough to demonstrate the overall DIA spectral quality for unambiguous site localization.

Response: In this study, we designed several strategies to improve the identification and site localization confidence. (1) We aimed to maintain uniformity in peptide retention features and fragmentation profiles in our large-scale database; all DDA and DIA acquisition were performed using similar chromatographic gradient and acquisition parameters in Orbitrap Lumos MS, and spiking indexed retention time (iRT) peptides. (2) We utilized stringent phosphopeptide analysis in two levels: (A) database search with 1% false-discovery rate (FDR) at PSM, protein level; and (B) criteria of Class 1 site localization (probability of 0.75 or higher). Bekker-Jensen et. al. has reported an algorithm for site localization probability estimation which has been integrated into the commercial Spectronaut software (*Nat. Commun* 2020). Thus, we adapted Spectronaut in the spectral library construction and DIA data analysis for site localization determination using the new version of Spectronaut. Using the new Spectronaut (V14) search for phosphoproteome datasets (n=180 raw files), 121,407 (76%) among 159,524 phosphopeptides have class 1 localized sites in our hybrid library (**Fig. 3b**).

Following the comment, new **Fig. 3b-c** were revised and descriptions were added to the manuscript.

Page 10, line 223-230: *“Finally, hybrid phosphoproteome library was generated combining fractionated DIA (n=24 raw files) with the DDA (n=156 raw files) datasets using Spectronaut Pulsar search at 1% FDR at (PSM, peptide and protein levels) (Mol Omics 2019, 15, 348-360). Taken together, we constructed a library consisting of 159,524 phosphopeptides (203,550 precursors corresponding to 88,107 phosphosites on 8,805 protein groups (Fig. 3a, see details in Supplementary Table 3). With phosphosite localization, this library includes 121,407 class 1 (≥ 0.75 probability) phosphopeptides (Fig. 3b) indicating that majority have highly accurate site localization.”*

4. In page 8, for the coverage of the phosphotyrosine-enhanced hybrid library (Fig.3d-f), how many class 1 sites were identified in these pathways? Probability score ≥ 0.75 is a general criterion

to dissect the biological function of a phosphorylation site from a phosphoproteome data. It remains unclear how many key phosphosites were identified in the RTK/RAS/RAF pathways and other targetable proteins. This is critical and helpful to understand lung cancer mechanism and identify therapeutic targets.

Response: Thanks for the comments. We fully agree that a highly confident criterion has to be applied to dissect the biological function of a phosphorylation site.

4.1 In our hybrid library, 16,281 phosphosites from 1,329 protein groups were annotated as **cancer related pathways (Fig. 3d)** and as high as 85% (13,862) were class 1 localized sites (≥ 0.75), including ErbB signaling (921 phosphosites, 800 class 1 on 65 proteins, 78%), NSCLC signaling (568 sites, 471 class 1 from 52 proteins, 79%), and the EGFR-tyrosine kinase inhibitor (TKI) resistance pathway (743 sites, 648 class 1 from 59 proteins, 76%). For **385 kinases** and **140 phosphatases (Fig. 3e)**, 85% among 5,091 phosphosites (4268 class 1) and 1429 phosphosites (1226 class 1) were Class 1 sites, respectively (**Supplementary Table 3**).

4.2 For the **RTK/RAS/RAF pathway**, 169 phosphosites are covered and 101 are class 1 sites (**Supplementary Fig. 4a**). These include crucial phosphosites and 20 newly identified sites.

Following the comment, we added **Supplementary Fig. 3** to show these class 1 sites in the EGFR-RAS-RAF pathway. The following text is also added to the manuscript:

Page 12, line 263-273: *"In the example of EGFR-RAS-RAF pathway which is most crucial for lung cancer progression, 101 among 169 phosphosites are class 1 phosphosites (**Supplementary Fig. 4a**). Others includes known kinases such as RAF1-S642, BRAF-S365 and BRAF-T753 which are known to be phosphorylated by ERK2 kinase as well as MEK1/MEK2 substrate site (ERK1-T202, ERK1-T204, ERK2-T185 and ERK2-Y187) and RAF substrate site (MEK2-S222). In addition, EGFR-S1036, SOS1-S232, SOS1-1205 are among the newly identified class 1 sites. For the oncogene EGFR, the 55 phosphosites (40 class 1 sites) locating in the tyrosine kinases and autophosphorylation domain also include the Y1197 and Y1172 phosphosites, which are the characteristic autophosphorylation sites upon activating driver mutation of EGFR (**Supplementary Fig. 4b**).*

Supplementary Figure 4

Supplementary Figure 4. RTK-RAS/RAF pathway phosphosite coverage. Phosphosites (class 1) in EGFR/RAS/RAF pathway. **(a)** Phosphosites covered in library from RTK-RAS/RAF pathway. Among total of 169 identified sites, 101 were class 1 including 14 tyrosine sites. **(b)** Phosphosite coverage (class 1) of EGFR. Among 55 phosphosites identified, 40 were localized as class 1 including 12 tyrosine phosphorylation.

5. In page 8, The authors found only 22.6% phosphopeptide overlap between cell lines and NSCLC tumor samples. It was somewhat surprising. Why the overlap is so low between the samples with tissue origin?

Response: Thanks for the comment. We would like to clarify the description. As shown in the following Venn diagram, 54% (n=20,927) of tissue phosphopeptides were commonly identified in the fractionation datasets from the cell lines, which is likely due to the tissue-derived or enriched proteins. As a comparison, the overlapping identification of two cell lines is about 60%. We have corrected the description in the main text as follows.

Page 10, line 216-219: *"...The results indicate that tumor tissues provide additional 17,817 phosphopeptides (44%) in addition to 56% commonly observed in the fractionation dataset from the cell lines, likely due to the tissue-derived or enriched proteins in tumor samples."*

Figure. Overlapping phosphopeptides identified between (a) tissue and cell lines, and (b) PC9 cells and CL68 cells.

6. In page 11, the authors constructed the database by using the samples only originating from NSCLC (cell lines and patients), but they further evaluated its performance in breast cancer. I do not see the value of this experiment. As I mentioned in the above comment, even the phosphopeptide overlap between samples from the same tissue origin is so low. The tissue-specific phosphosite loss could be huge. Can the authors justify the value of this strategy for phosphoproteome analysis of other types of cancer?

Response: In this study we constructed a reference library from lung cancer samples and tried to demonstrate its general utility for other human samples, such as breast cancer tissues. There are several justification behind this experiment. (1) According to the datasets of 44 normal tissues of different organs in the Human Proteome Atlas (HPA), only 4% among 19, 670 proteins were classified as tissue-specific proteins. The breast and lung tissues possess as high as 90% common proteins (**Fig. a**, below). (2) Comparing our in-house breast cancer proteome library constructed from tissue and cell lines (10,652 proteins, unpublished) and our lung cancer proteome library reported in this study, 97 % protein groups were commonly present. (**Fig. b**, below). The single-shot DIA analysis results of our breast cancer cell line (MDA-MB 453) also showed significant common coverage (66%) compared to that of lung cancer cell line (CL68)(**Fig. c**, below). Given the high percentage of similar human tissue proteome composition, we expected that the constructed proteome and phosphoproteome library and integrated single-shot DIA strategy will provide a useful resource to advance phosphoproteomics applications for diverse sample types.

Figure. Comparison of protein and peptide composition between lung and breast cancer tissue. (a) Proteins detected by immunohistochemistry in normal breast and lung cancer tissue from Human Proteome Atlas (HPA). (b) Overlap of protein groups of spectral library constructed from lung cancer and breast cancer. (c) Phosphopeptides from triplicate DIA data in using breast cancer cell line (MDA-MB-453) and lung cancer cell line (CL68).

We have discussed this issue in the revised manuscript.

Page 15, line 343-347: *“In this study we constructed a reference library from lung cancer samples and further demonstrated enhanced profiling coverage and quantitation performance. Given the high percentage of similar proteome composition within human tissue, the general utility of GPS for diverse sample types was evaluated using other human specimens to evaluate its application.”*

7. The main conclusion of “single-shot DIA offers highly reproducible large-scale phosphosite quantification” section is that libDIA could achieve higher phosphosite identification and higher quantification reproducibility than DDA method. These conclusions are well-known merits of DIA method itself. What is the advantage/disadvantage of GPS strategy over other DIA methods? This question should be clearly addressed in this study.

Response: Thank you for the comment. Although the performance of DIA for proteome profiling had been nicely demonstrated by several groups, only limited studies reported DIA approach for the more challenging PTM characterization and especially large scale phosphoproteome study were not available. To the best of our knowledge, a very recent study by Olsen’s group is the only global phosphoproteomics to report quantification of > 29,000 phosphopeptides (~14,000 localized phosphosites) by a fast LC and DIA method (*Nat. Commun* 2020). In their study, direct DIA approach was recommended for ease of application and access to wider community as library construction is demanding. (1) Compared to the previous DIA approach, our results demonstrated that a rich and confident phosphopeptide spectra library facilitates highly sensitive DIA-based phosphoproteome profiling of small amounts of samples including cancerous tissues. (2) On the informatic aspect, the integrated *dirDIA* and *libDIA* pipeline reveals complementary profiling results likely due to different data deconvolution. (3) Most importantly, our results showed that *libDIA* significantly outperforms all methods. The result of **95% phosphosites covered by libDIA** revealed the strength of the spectrum-centric approach to map large-scale reference libraries. (4) Finally, we also provided a comprehensive protein level library, which can be used to normalize the protein expression for precise interpretation of phosphorylation stoichiometry. Therefore the deep resource of over 88,000 phosphosites library and over 12,000 protein groups library will be useful resources for DIA-based quantification.

8. In page 13, the section of phosphoproteomics profiling of EGFR-TKI-sensitive and resistant lung cancer cells is largely descriptive. The result of this section does not provide a clear methodology advancement or new mechanistic insight in this type of study. The experimental design remains questionable, either. There are many different mechanisms for intrinsic/acquired resistance for EGFR inhibitors, such as EGFR target alterations, bypass tracks (MET amplification, HER2 amplification, NF1 loss, etc.). Why the authors chose to compare two cell lines with different genetic backgrounds? What is the known resistant mechanism for CL68 cell? What is the new information that GPS strategy could provide? Does the high phosphotyrosine proteome coverage of this strategy contribute to the new mechanistic understanding of TKI resistance?

Response:

8.1 Choice of cell line: Despite the efficacy of targeted therapy using tyrosine kinase inhibitor (TKI) for patients with activating driver mutation on EGFR (major on Del19 and L858R), management of patients with EGFR mutations who eventually develop resistance to TKI has become the biggest challenge in lung cancer therapy. In East Asia, the primary cause of resistance

to TKI is driven, in approximately 60% of advanced lung cancer patients, by acquiring an additional EGFR T790M point mutation located at the gatekeeper position of the ATP binding site. To reveal the molecular landscape of acquired resistance, in this study, thus, we characterized pairs of cell lines with primary activating mutation and acquired secondary mutation on EGFR. Due to the lack of paired cell lines from the same parental cells, we chose PC9 cells which has EGFR Exon-19 deletion and is sensitive to TKI (10~30 nM), while CL68 cells possess a double mutation of exon-19 deletion and T790M point mutation after Iressa and chemotherapy treatment, and has no sensitivity to TKI (IC₅₀ >10uM) [Yen et al., *Proc Natl Acad Sci U S A* 2015, **112**, 6955-6960]. We have added description in the revised manuscript:

Page 17, line 380-386: *“In East Asia, the primary cause of resistance to TKI is driven, in approximately 60% of advanced lung cancer patients, by acquiring an additional EGFR T790M point mutation located at the gatekeeper position of the ATP binding site (Pao et al., PLoS Med. 2005, 2, e73). We applied the GPS approach to quantitatively compare TKI-sensitive PC9 with exon-19 deletion (IC₅₀=30 nM) and TKI-resistant CL68 (IC₅₀=20 μM) cells with a double mutation of exon-19 deletion and T790M point mutation after Iressa and chemotherapy treatment, which may provide insight into the drug resistance mechanism (Yen et al., Proc Natl Acad Sci 2015).”*

8.2 *Differential comparison between the TKI-sensitive PC9 and TKI-resistant CL68 cells result in 747 upregulated and 1011 downregulated phosphosites in resistant cells compared to phosphosites in sensitive cells (two-sample t-test, S0=0.1, FDR<0.01). These include NSCLC signaling, endocytosis, ErbB signaling, the EGFR-TKI resistance pathway, and Ras signaling in KEGG which have been reported to be associated with TKI resistance in NSCLC. Several cancer-associated pathways, such as adherens junctions, tight junctions, and focal adhesions associated with epithelial–mesenchymal transition (EMT), were also enriched (Fig. 5c). EMT has been reported as a major hallmark of EGFR-TKI resistance in NSCLC (Zhu et al., Front. Oncol. 2019, 9, 1044-1059). Our results may reveal elevated site-specific phosphorylation in an EMT event. The EGFR-TKI resistance pathway is among the most deregulated pathways (p <0.05), 161 phosphosites covering almost all downstream proteins were observed (Fig. 5e), and 22 phosphosites showed differential levels, including the higher phosphorylation level of a autophosphorylation site (Y1197 and Y1172) and kinase domain (Y727) on EGFR, accompanied by protein overexpression at the PI3K/Akt and SRC/STAT3 subpathways.*

“The dataset also facilitate the identification of the upstream kinases responsible for TKI resistance, including top ranking PKA (903 substrates), PKC (858 substrates), ERK 1/2 kinase motif (755 substrates) (Fig. 5f-g) as well as in tyrosine kinases Src with their overexpressed substrates (n=18) in the TKI-resistant CL68 cells. In addition, observed upregulation of the TPX2-S121 and S125 sites likely correlated with the reported role of AURKA kinase and its coactivator TPX2 in response to chronic EGFR inhibition to mitigate drug-induced apoptosis in resistant lung cancer cells (Shah et al., Nat. Med. 2019, 25, 111-118). We further quantitatively compared the alterations in 646 pTyr sites, of which 43 sites showed differential phosphorylation (Fig. 5h). Many upregulated sites are associated with EGFR (Y1197, Y1172 and Y727) and its adaptor proteins DLG3-Y673, likely due to EGFR-activating mutations that drive its downstream signaling

cascade. Whether these identified kinases and phosphosites may confer the transformation from TKI-tolerant to TKI-resistant cells remains to be validated.”

9. In page 15, the questions remain similar for phosphoproteome profiling in lung cancer tissues. The result is largely descriptive. The experimental design remains unclear. The authors chose 3 early-stage tumors and 2 late-stage tumors. Lung cancer is of high heterogeneity. There are so many factors that impact lung cancer pathogenesis, such as cancer driver genes, age, gender, smoking status, environment, etc. Such a small number of samples could be very difficult to yield biological insight. Again, what is the new information or advancement that GPS strategy could provide for clinical sample analysis? The authors published a large-scale TMT-based proteogenomics study of lung adenocarcinoma clinical samples (*Cell*. 2020,182: 226-244). What is the advantage of GPS strategy over the TMT-based approach for clinical sample analysis? A detailed comparison between these two studies is necessary.

Response: Thanks for the comments to clarify if the reported strategy has provided new insight on the cancer biology. Aiming to provide technical advancement on the phosphoproteomics profiling, the main objective in this study is to demonstrate the analytical merits of DIA-based approach using cell line and tissue and their potential for large-scale analysis of clinical samples. Following this comment, we have summarized the comparison on the results of 102 pairs of tissues obtained from TMT-labeled (*Cell* 2020) and single-shot DIA phosphoproteome approach as follows:

9.1 Our DIA approach provides profiling of lower missing values compared to TMT strategy. Triplicate analysis of 10-plex TMT labeled peptides by DDA approach quantified 20,000 class 1 phosphosites (31,428 phosphopeptides). However, only 11,277 (36%) were quantified in all the three batches. From combined data sets of tissue samples (*Cell* 2020), only 3.7% of 166,792 phosphopeptides were reproducibly detected in at least 75% of the samples (>0.2 mg) in 80 patients (by 12 months of nonstop data acquisition); a large amount of phosphoproteomics data were generated with very limited utility. As a comparison, significantly low between-run missing values of 2.5% and 0.2% were observed in *libDIA* and *dirDIA*, respectively,

9.2 DIA provides superior quantification accuracy. Although comparison of TMT with DIA was not the scope of this manuscript, our analysis of tissue phosphoproteome demonstrated superior quantification of DIA to improve the known problem of ratio compression (tumor-to-normal). The following table compares the phosphosites ratio for 4 pairs of tissue samples obtained by DIA and TMT approaches. The DIA has generally larger degree of ratio compared to TMT. For example, EGFR-T693 known to have decreased expression in lung cancer has more obvious down-regulation in DIA result. In summary, DIA based label free approach may be useful for large scale discovery or verification in tissues.

Gene	Position	1260T/N		1244T/N		1326T/N		1332T/N		1259T/N	
		TMT	DIA	TMT	DIA	TMT	DIA	TMT	DIA	TMT	DIA
BRAF	S446	0.82	1.26	0.90	1.30	0.71	1.48	1.25	2.53	2.20	1.45
EGFR	T693	0.89	0.51	0.68	0.15	0.32	0.22	0.95	0.41	0.56	0.42
PML	S403	1.03	2.13	1.23	0.41	1.34	1.64	0.70	0.75	1.22	1.65
CTND1	S252	0.98	0.50	0.90	0.85	0.78	0.77	0.82	0.49	1.60	1.72
ERBB3	S686	1.49	1.28	0.54	0.22	0.56	0.74	1.10	0.80	0.72	0.39
ERBB3	S982	1.15	0.85	0.41	0.11	0.54	0.47	0.84	0.39	0.80	0.31
PML	S518	0.92	0.66	0.93	1.22	1.04	1.24	0.85	0.98	1.88	2.29
PML	T409	1.28	5.89	1.00	1.18	1.06	1.28	0.85	1.60	1.01	0.08

Figure. Comparison of tumor/normal phosphosite ratio between TMT and DIA approach.

Red shows upregulation (cutoff 1.33 in TMT and 2.0 in DIA) while green shows down regulation (cut off 0.77 in TMT and 0.5 in DIA).

9.3 Highly sensitive phosphoproteomics with deep coverage comparable to fractionation TMT data, In the current large scale tissue analysis, TMT approach usually required a large amount of starting peptide (~2mg) for deep profiling. With the high number of fractions, days of data acquisition and downstream analysis are required. By using DIA approach in this study, single-shot analysis (triplicate runs) provides a phosphosite identification coverage comparable to the results by a StageTip-based fractionation (7 fractions, duplicate runs). Therefore DIA quantification demonstrated high sensitivity, deep coverage and analysis efficiency that can be applied to clinical samples with minute sample amounts.

In summary, compared to the TMT-based strategy, our single-shot DIA analysis demonstrated strengths in **high sensitivity, deep coverage, high efficiency, low missing values** as well as **resolving the ratio compression problem.**

Minor concerns:

1. The authors collected 25 tumor samples, but only 22 of them were used for spectral library construction. The selection criterion need to be provided.

Response: All the 25 tumor tissues were used for either library construction or DIA analysis.(1) To construct a library towards comprehensive proteome coverage, diversity of tissue samples were selected from EGFR mutation status (Del19, L858R, WT), and clinical staging on early (I-II) and late (III-VI)) stages. In addition, a pooled sample strategy was used to increase proteome/phosphoproteome coverage by patient heterogeneity. (2) To demonstrate DIA-based tissue profiling, tumors from early (n=3) and late stage (n=2) with Wild type and Exon-19 deletion mutation were used for singleshot DIA analysis. Due to the small size of early stage tissue (IA), 3 tumors from Stage IA and IB were only used for singleshot DIA-based proteome and phosphoproteome analysis. The detailed information can be found in **Supplementary Table 1b.**

2. The authors cultured five cell lines in the Methods section but only four were used for database construction. The selection criterion need to be provided.

Response: Thanks for the comment. For construction of the proteome library, all the 5 cell lines were used. For construction of phosphoproteome library, 4 cell lines with two major driver EGFR mutation as well as the acquired mutation T790M were used, including PC9 (Exon-19 deletion), CL68 (Exon-19 deletion and T790M), H3255 (L858R) and H1975 (L858R and T790M) (**Supplementary Table 1**).

3. In Fig 2A, b9-98 peak could not be found on the MS/MS spectra of the modified peptides with phosphorylation in Y1172.

Response: Thanks for pointing out the point. We have added the annotation of b9-98 peak in the **Fig. 2a** (shown below).

Figure 2

4. What is the criterion to define up-regulated or down-regulated phosphosites in the last two sections?

Response: For cell line experiment (**Fig. 5**), the criteria used was adjusted FDR of <0.01, $S_0=0.1$ from two-sample t-test performed using Perseus. For tissue phosphoproteome profiling, the data after normalization to the protein expression level were computed to determine the differential ratios (p-value <0.05 and $S_0=0.1$).

5. The authors collected cells and heated them at 95 °C for 5 minutes. Did the author consider effect of high temperature on phosphorylation status? In addition, the authors chose 29 °C for protein alkylation and trypsin digestion, which was an unusual reaction condition. The authors have to justify such procedure.

Response: We described the rationale for the protocol as follows.

5.1 Heat treatment (5 min at 95 °C) upon lysis of cell and tissue samples (Step 2) is used to inactivate the endogenous proteases and phosphatases (*Nat Protoc* 2018, **13**, 1897-1916), which is useful to improve phosphoproteome identification. The strategy has been reported by Svensson et al. (*J Proteome Res.* 2009, **8**, 974-981) and also used previously for 14 rat organs and

tissue samples covering 31,480 phosphorylation sites by Olsen's group (*Nat. Commun.* 2012,**3**, 876). Our group also used this protocol to report large scale human tissue analysis (*Cell* 2020, **182**, 226-244.e217).

5.2 Urea is widely used to aid in protein denaturation and solubilization. However carbamylation at the N-termini of proteins/peptides and at the side chain amino groups of lysine and arginine residues affects digestion and protein identification/quantification. The use of urea is recommended for 25 °C (*Nat. Protoc.* 2018, **13**, 1632-1661). Betancourt et al., demonstrated room temperature achieved higher digestion efficiency and lower carbamylation (*J Proteome Res.* 2018, **17**, 2556-2561), which was also used in large scale study (*Cell* 2020, **182**, 200-225 e235). Recent work from our group on large-scale tissue proteome and phosphoproteome also employed 29°C for reduction/alkylation and digestion (*Cell* 2020, **182**, 226-244.e217). Therefore, we adopted this strategy.

Following the comments we also add the rational with references in the revised manuscript for better clarity

Page 26, line 588-590: *"The homogenized samples were heated at 95°C with vortexing at 750 rpm for 5 min to inactivate the endogenous proteases and phosphatases (Nat Protoc 2018, 13, 1897-1916) and sonicated for 10 min (30 s on, 30 s off) using Bioruptor Plus (Diagenode, Denville, NJ)."*

Page 27, line 614-617: *"The extracted proteins were re-suspended in 8 M Urea, then reduced by 10 mM dithiothreitol at 29°C for 30 min and alkylated with 50 mM iodoacetamide at 29°C for 30 min in the dark with temperature selection aimed at reducing urea carbamylation (J Proteome Res. 2018, 17, 2556-2561, Cell 2020)."*

6. The authors used both Phosphosite plus and UniProtKB for the analysis of their data, but the criterion for these two databases was not clearly presented.

Response: The PhosphositePlus database deposits experimentally observed with high confidence of manually curated protein modification sites from over 22 000 articles and thousands of MS datasets (Hornbeck et al., 2018, PMID: 30445427). Only sites with probability scores of $P \geq 0.95$ or Ascore ≥ 13 were collected. . In addition to annotation information from literature, the UniProt Knowledgebase (UniProtKB) also contains computationally analyzed records, which have not been experimentally verified. Accordingly, we compared our data to the PhosphositePlus database following the same criteria of probability scores of $P \geq 0.95$ or Ascore ≥ 13 .

Reviewer #2 (Remarks to the Author):

The authors Kitata et al describe the development of a large lung cancer-specific phosphopeptide library and apply it to measure phosphorylation signaling in lung cancer cell lines using DIA. In general, the manuscript is well written and well reasoned. I have some concerns about how localization was performed with PhosphoRS, given that the tool was originally designed for DDA.

In particular, I am somewhat concerned that fragment ion reports generated by most DIA software do not include co-eluting fragments (and noise) that were not assigned to the "detected" peptide. These fragments and noise are crucial for the PhosphoRS localization algorithm to perform correctly, since it is based on the binomial probability of randomly finding fragment ions given the background ion frequency. In particular, co-eluting alternate localization forms will generate localization-important ions that are not considered in these reports. The absence of these ions will provide a false sense of security that the "detected" peptide has no other competing alternate localization possibilities. As a result, the localization probabilities generated by PhosphoRS may appear accurate in the control experiment with only a few, known phosphopeptides in a background of unphosphorylated yeast peptides, but not actually be accurate in the presence of a complex phosphopeptide matrix with more co-eluting phosphopeptides.

In addition, the authors do not discuss quantification using only site localizing ions, which may complicate the measurements of co-eluting phosphopeptide positional isomers. Both of these concerns would be somewhat mitigated if the authors demonstrated the fraction of localized phosphopeptides that have potentially overlapping elution patterns using their methods.

Response: Thank you for the comments to improve our manuscript. Based on the suggestions from all Reviewers, we have performed the following major revisions to address the comments on the site localization in the library as well as DIA datasets. (Please also see the Summary in the beginning of this point-by-point reply letter).

(1) **Reconstruction of spectral library to improve stringency.** By using new Spectronaut (V14), the hybrid phosphoproteome library established using DDA and DIA datasets has been reconstructed with 1% FDR (PSM, peptide and protein) and site localization using a reported strategy (Muntel et al. *Mol Omics* 2019 PMID: 31465043). In brief, the DDA (n=156) raw files were first searched by Spectronaut Pulsar and stored as archive. Then the fractionated DIA (n=24 raw files) were processed by Spectronaut Pulsar and further combined with the DDA search archive with overall FDR cut off to construct the spectral library. The new library consists of 159,524 phosphopeptides (121,407 pass class 1 phosphosite localization probability) corresponding to 88,107 phosphosites. ALL the DIA datasets in this study were re-analyzed using the reconstructed library accordingly. The whole manuscript has been extensively revised accordingly.

(2) **Determination of Phosphosite Localization Probability.** When the manuscript was prepared (Spectronaut version 12), the available DIA tools did not provide phosphosite localization. To

enhance the site localization confidence, we adapted site localization tools using the rich fragments peak lists generated by DIA method in accordance with the PhosphoRS which was commonly used for DDA dataset [Taus et al., *J. Proteome Res.* 2011]. Following the approach of PhosphoRS, in the previous manuscript, the fragment features including mass accuracy and peak area obtained from DIA spectra were used to calculate the site localization probability to distinguish phosphopeptide isomers. The new version of Spectronaut (version 13 onwards) was recently launched and added the phosphosite localization probability calculation (Bekker-Jensen et al., *Nat. Commun.*, 2020). In the revised manuscript, we have adapted Spectronaut (V14) to reanalyze the site localization confidence for all the datasets.

A panel of 166 synthetic phosphopeptides were used to evaluate the site localization confidence. We also reconstructed synthetic phosphopeptide library (166 phosphopeptides) and re-analyzed the site localization in Spectronaut. Using the reconstructed library, the new results indicated that 96% of the 166 phosphosites obtained from DIA are class 1 localized.

Finally, I have the following minor comments:

1) Page 4, paragraph 1: This section on phosphoproteomics with DIA should cite Rosenberger et al 2017 (PMID: 28604659) and Meyer et al 2017 (PMID: 28661500) as previous efforts to analyze and site localize phosphopeptides. In addition, this section should also cite Lawrence et al 2016 (PMID: 27018578), which demonstrated the construction of a similar large-scale phosphopeptide library with applications in DIA.

Response: Thank you for adding references of previous work. We included the citations of Meyer et al., *Nat Methods* 2017; Rosenberger et al., *Nat Biotech* 2017; Lawrence et al., *Nat Methods* 2016; and other groups' reports to introduce their effort for phosphosite localization and generation of phosphoproteome databases.

Page 4, line 85-87: *“Recently, to address the challenge in site-specific analysis of post-translational modification dataset, various algorithms have been introduced including Inference of Peptidoforms (Rosenberger et al., Nat Biotech 2017), Thesaurus (Searle et al., Nature Methods 2019), and PIQED (Meyer et al., Nat Methods 2017).*

Page 4, line 82-83: *“Lawrence et al., also reported a large-scale phosphoproteome database as a resource for targeted phosphosite quantification (Nat. Methods 2016).”*

2) Page 6, line 128: The methods section discusses library creation in more detail, but it would be beneficial to mention here that the MaxQuant DDA library was generated with Global FDR filtering. Interestingly, the methods section does not discuss site-level FDR but it is mentioned briefly here. The methods should reflect this filtering as well.

Response: Thanks for the suggestion. We have revised the description of MaxQuant analysis to add information on both global and site-level FDR of 1%. We have also added the details in the Result and Methods.

Page 14, line 322-325 (Results). *“The single-shot dirDIA and libDIA results were obtained from Spectronaut and compared to the results from single-shot DDA and peptide-fractionated DDA data were processed by MaxQuant label-free quantification (LFQ) with 1% FDR at site, PSM and protein level (Fig. 4a-b).”*

Page 32, line 719-720 (Methods): *“Protein and peptide were both filtered at global 1% false-discovery rate (FDR) at PSM and protein levels as well as phosphosite-level FDR.”*

3) Page 6, line 133: The tool used for DIA analysis (Spectronaut) should be mentioned here, because this citation (16) suggests that DIA-Umpire was used.

Response: Thanks for the comment. In the revised manuscript, we have corrected the citation to introduce Spectronaut by Bruderer et al. (*Mol. Cell. Proteomics* 2015, Ref. number 24).

4) Page 6, line 139: More details should be added about how PhosphoRS was adapted for DIA.

Response: Thanks for the comment. As we described earlier in this letter, we have adapted the phosphosite localization in the updated Spectronaut version (Version 14, Bekker-Jensen et al., *Nat. Commun.*, 2020). The details on the use of Spectronaut have been added in the Result and Methods.

“Page 7, line 151-155 (Result): “The high complexity of DIA signals poses a challenge for unambiguous site-specific phosphopeptide quantification. In the first step of confident phosphosite determination for each precursor, the phosphosite localization tool recently reported by Bekker-Jensen et al. (Nat. Commun 2020) and integrated into Spectronaut was applied to filter class 1 phosphosite localization.”

Page 35, line 786-788 (Method): *“Phosphosite localization tool recently and integrated into Spectronaut was applied to filter class 1 phosphosite localization and a class 1 probability cutoff ≥ 0.75 for DDA and libDIA and ≥ 0.99 for dirDIA employed (Bekker-Jensen et al. Nat. Commun 2020).”*

5) Page 8, line 170: Should "predicted retention times" be "library retention times"? The "predicted" language may confuse people into believing that the retention times were derived de novo (e.g. with SSRCalc or ProSit).

Response: Thanks for the suggestion. We have replaced it with “library-annotated retention time” in both Figure 2e and the text (Page 9, line 198).

6) Page 8, line 186: This should indicate which cell lines were used. Were they the same as PC9 and CL68 from later in that paragraph?

Response: Thanks for the comment. The information on the two cell lines, PC9 and CL68 cell lines, were added accordingly (Page 10, line 222).

7) Page 11, line 236: Was PhosphoRS used to localize peptides for both DIA and DDA, or was the built in MaxQuant localization algorithm used? If so, are these comparable since they produce very different types of scores?

Response: Thanks for raising the issue. For singleshot DDA data, we used MaxQuant software to obtain both LFQ quantification results and site localization probability. As commented, using similar software will enable more unbiased comparison. With the incorporation of Spectronaut based localization (Bekker-Jensen et al., *Nat. Commun.* 2020), reported getting equivalent result with the site localization algorithm implemented in MaxQuant.

8) Page 11, line 249: This section indicates that both direct and library searches were combined to identify phosphorites. Were they FDR corrected together afterwards, or is the actual FDR inflated from errors in both searches? How are the search and localization scores comparable since they were derived using different types of data (e.g. the DIA reports may have no alternate isomer localization ions [see above] or noise, while the DDA reports have both noise and alternate ions)?

Response: Thanks for the comment. For reporting singleshot DIA analysis results processed by direct DIA (*dirDIA*) and library based DIA (*libDIA*), we show both independent results and their overall quantitation results to show their complementary processing nature (**Figure 4a-b**); it is noted that > 95% phosphosites of overall results were covered by *libDIA* approach. As commented, consistent FDR correction is very critical yet difficult to obtain with the existing DIA analysis platform and output report. To avoid the biased FDR issue that the Reviewer pointed out, library based DIA analysis was used throughout all other DIA datasets of the manuscript.

9) Page 14, line 319: The motif enrichment tool that was used should be indicated and cited.

Response: Thanks for the comment. Motif enrichment was performed by using the PhosphositePlus tool and Perseus. To visualize the amino acid composition of these motifs, the motif logo was generated using pLOGO (O'Shea et al., *Nat Methods* 2013) and phosphorylation motif enrichment was performed based on Fisher exact test (FDR<0.02). We have added the information in both Result (Page 18, line 412-413) and Methods (Page 35, line 802-806) of the revised manuscript

10) Page 23, paragraphs 2 and 3: Two types of peptide fractionation were performed: stagetip fractionation and fractionation with an HPLC. The samples that were prepared with each type of fractionation should be indicated. Additionally, these sections should be more clearly delineated so that it's clear these are two types approaches.

Response: Thanks for the comment. Reversed-phase peptide fractionation was performed for tryptic peptides from cell lines and tissues using the StageTip format and HPLC column format, respectively. We have revised the description in the Result and Methods as shown below.

Page 6, line 131-134 (Result): "...(2) using high pH reversed-phase (HpRP) chromatography for fractionation of tryptic peptides from pooled tissue (column) [Chen et al., *Cell*, 2020] and individual cell lysate (StageTip)[Dimayacyac-Esleta et al, *Anal. Chem.* 2015], followed by

phosphopeptide enrichment using immobilized metal affinity chromatography (IMAC) in the StageTip protocol;...”

Page 28, line 627-633 (Methods): *“The StageTip was prepared by packing reversed-phase membranes styrene divinylbenzene resin modified with sulfonic acid groups (SDB-RPS) membranes Empore™ disks into the Gilson 200-μL tips. For peptide fractionation from cell lysate, tryptic peptides obtained by in-StageTip digestion were fractionated and eluted from reversed-phase Stage-Tip using buffers with increasing ACN percentage (10%, 15%, 20%, 30%, 45%, 60% and 80%) prepared in 40 mM ammonium formate.”*

Page 28, line 639-643: *“For peptide fractionation from pooled tissue sample, tryptic peptides were re-dissolved in 0.6mL 5mM ammonium formate (pH 10) and 2% ACN, and loaded on a 4.6 mm x 250 mm Zorbax 300 Å Extend-C18 column (Agilent, 3.5 μm bead size) at a flow rate of 0.5 mL/min on a Waters alliance e2695 HPLC instrument coupled with Waters 2489 UV/Visible detector and fraction collector III...”*

11) Page 24, line 549: How were the synthetic peptides made? Where they purchased? From whom?

Response: The synthetic phosphopeptides were purchased from Synpeptide Co. Ltd (Shanghai, China). The company information was already provided in the “Chemicals and Materials” section (page 24, line 550-551).

12) Page 26, line 580: The MS/MS maximum injection time setting should be indicated for the DIA method.

Response: Thanks for the comment. For MS/MS scan at resolution of 30,000, the expected maximum injection time is 54 ms. We have added the information in Method (Page 34, line 774-775).

13) Supplementary Note, page 6: The discussion of the PhosphoRS reimplementation indicates the use of phospho neutral losses (e.g. -98 Da) for localization. These ions are generally considered untrustworthy for localization because they can also be generated from unphosphorylated series and theonines through commonly occurring loss of water (-80,-18).

Response: Thanks for the comments. We agree that the role of neutral losses fragment ions should be evaluated when applying site localization. In this study, we also evaluated the result using different numbers of fragment ions, including after removing neutral loss ions (annotated as top30 noNL) in the previous version of the manuscript. As shown in **Figure** (shown below), the number of quantified phosphopeptides is lower compared to the results using total precursors or Top 45, top 30 and top 15 precursors. However, we finally used localization strategy by Spectronaut in the revised version.

Figure. Comparison of number of quantified phosphopeptide precursors using different number of fragments per phosphopeptide precursor in the spectra library, including top 15, top 30, top 45, all fragments, and those without neutral loss (noNL) fragment ions.

Reviewer #3 (Remarks to the Author):

== Conclusion ==

In their manuscript submission, Kitata et al. describe a comprehensive MS DIA strategy to both acquire and process deep profiling phospho-proteomics data. They combine a well-established and robust phospho-proteomics workflow with novel approaches in data analysis (hybrid DDA + DIA spectral library; hybrid dirDIA + libDIA analysis; phosphoRS phospho-site-localization for DIA data) to generate the deepest phospho-proteomics spectral library for a single cancer type. This is an impressive and extensive amount of work.

From my perspective, the biggest strength of the manuscript is its deep, high quality phospho-proteomics spectral library, acquired by samples generated and measured by the same group on a single type of MS instrument. High quality phospho-proteomics spectral libraries cannot yet reliably be generated in silico such as proteome spectral libraries, and as such are rare and have many uses in the community. Furthermore, it seems to be the deepest generated for a single human cancer type (NSCLC). At the same time, this may also limit its most comprehensive coverage to NSCLC samples, as phospho-peptides from other cancers / tissues may not be represented. For these types of samples, a broad coverage library such as the “plug-and-play” library generated by Lawrence et al. (2016), which covers 109,611 phosphorylation sites from a variety of cell types, might be more applicable.

On the data analysis side, the adaptation of the publicly available phosphoRS localization tool to DIA input data further provides a good alternative to the already available DIA PTM localization solutions Thesaurus (Searle 2019) and Spectronaut (Bekker-Jensen 2020). The novel ways to generate DDA/DIA hybrid spectral libraries and merge the output of Spectronaut dirDIA and libDIA I find intriguing, but have a few concerns pointed out in the comments below.

All in all, I believe this study and the high-quality data presented in it will be of great value to the phospho-proteomics community. For this reason, I would recommend publication in Nature Communications with revision of the points below.

Response: Thank you for the kind comment supporting our manuscript. Following the comments, we have performed the following analysis and revisions. (Please also see the Summary in the beginning of this point-by-point reply letter).

(1) Reconstruction of hybrid phosphoproteome and proteome library

1.1 To obtain spectral library with identification FDR control, we have **reconstructed a hybrid phosphoproteome library of combined datasets** (n=180 raw files) using recently updated Spectronaut Pulsar v14 at 1% FDR cutoff (PSM, peptide and protein levels). This new library consists of 159,524 phosphopeptides (121,407 class 1) corresponding to 88,107 phosphosites on 8,805 protein groups. (**Figure 3b**). For confident phosphosite determination for each precursor, the phosphosite localization tool recently reported by Bekker-Jensen et al. (*Nat. Commun.* 2020) and integrated into Spectronaut was applied to filter class 1 phosphosite localization.

1.2 We also updated the proteome spectral library (n=191 DDA raw files) of >12,000 protein groups using Spectronaut v14. The performance of DIA for proteomic profiling in cell lines and tissues were added in **Supplementary Fig. 6 and 7**.

1.3 To show the strength of our hybrid library, we added analysis to compare the performance of single-shot DIA analysis using either hybrid phosphoproteome library or library constructed using a single type of dataset (DDA or DIA). The hybrid library enhances identification of phosphopeptides by 39% and 12% compared to DDA-based and DIA-based libraries, respectively (**Supplementary Fig. 3**).

(2) **Computation of phosphosite localization and validation using 166 Synthetic phosphopeptides** We incorporated the **phosphosite localization probability calculation** using the updated version of Spectronaut (Version 14, Bekker-Jensen et al., *Nat. Commun.*, 2020) and re-analyzed all the datasets. We also reconstructed synthetic phosphopeptide library (166 phosphopeptides) and used it to validate the site localization using Spectronaut (V14). The new results indicated that overall 96% are class 1 phosphosite. We further showed the potential of DIA for the confident identification, site localization and quantification of multiple phosphorylated peptides (**Supplementary Fig. 2**, Page 8, line 183-188).

(3) We have added **comparison of our GPS library with public large-scale phosphoproteome databases**, including PhosphositePlus, large-scale phosphoproteome from single cell type (Sharma et al., *Cell Rep.* 2017), Plug-and-play library (Lawrence et al., *Nat. Methods* 2016) and data from TMT based lung cancer tissue (Chen, et al., *Cell* 2020). The summary can be found in **Supplementary Fig. 5** (shown below). The revision is as follows:

Page 12, line 278-298: *“To the best of our knowledge, the 88,107 phosphosites in this library provide the deepest phosphoproteome coverage from a single cancer type over a single MS platform. Compared to the deepest phosphosite coverage from single cell type (HeLa) of 50,497 phosphosites reported by Sharma et al. (Cell Rep. 2014, 8, 1583-1594), our DDA data (64,962 sites) in the GPS library processed by the same search platform of MaxQuant still presented a 28.6% increase of phosphosites (Supplementary Fig. 5a). Compared to the PhosphoSitePlus database (239,180 phosphosites, as of October 23, 2020), our library contains 26,234 additional phosphosites (17,097 sites from hybrid class 1 localized library) (Supplementary Fig. 5b), including 1,589 newly identified tyrosine phosphorylation sites (Supplementary Table 3). Examples include novel sites with high localization probability ≥ 0.95 : EGFR-S1036, STAT3-T716, SRC-S212, ALK-S76/77/78, and PLCG2-Y13/S785, in the NSCLC pathway. The “plug-and-play” database (109,611 phosphosites, 11,428 proteins) was constructed on the datasets of 989 LC-*

MS/MS runs using phosphopeptides enriched with Fe-IMAC and TiO₂ from MCF7, HeLa S3 and HepG2 cell lines over different MS instruments (LTQ Orbitrap Velos, Elite, Fusion and Q-Exactive), which is a comprehensive resource to provide targeted assays in phosphoproteome analysis (Nat. Methods 2016, 13, 431–434). Compared to the “plug-and-play” database, 73,808 (46% of 159,524) phosphopeptides were in common with 85,716 phosphopeptides that are uniquely present in our GPS library (Supplementary Fig. 5c), suggesting the complementary nature of our GPS library.

Supplementary Figure 5

Supplementary Figure 5. Comparison of global phosphoproteome system library with large-scale phosphoproteomics reports. (a) Comparison of phosphosites identified from HeLa by Sharma et al. (Cell Rep 2014, 8, 1583-1594) and our datasets obtained by DDA-only library from GPS. **(b)** Comparison of phosphosites from PhosphositePlus database (2020/10/23 date) and GPS library. **(c)** Comparison of phosphopeptide from GPS library with Plug-and-play library by Lawrence et al. (Nat Methods 2016,13, 431-434)

== Major Comments ==

1. The authors state in their study that the generation of a hybrid spectral library (DDA + DIA runs) may provide gains in peptide identifications (IDs) compared to a single type library. Even though I think this is an intriguing hypothesis, no data is provided to compare advantages/disadvantages of a hybrid vs a single DDA or single DIA data. However, could the mix of DDA and DIA fragmentation patterns even be problematic, as it provides two different types of data during machine learning identification of peptides in DIA runs? Would it not be better to see if DDA or DIA runs are better for library generation, and the measure all library runs the same way?

Response: Thank you for the suggestions to further clarify the strength of hybrid library. Following the comment, we compared the performance of our hybrid library to the libraries constructed from the single DDA or DIA datasets. We have added the comparison results in **Supplementary Fig. 3** and revised the Result accordingly. The revisions are shown below.

Page 11, line 242-253: “The performance of the hybrid phosphoproteome library was evaluated by comparing to the spectra library constructed from a single acquisition type of DDA or DIA. Two datasets of comparable size were generated from DDA and DIA using PC9 cells coupled to StageTip fractionation (7 fractions) (Supplementary Fig. 3a). Three libraries were independently

constructed by DDA, DIA and hybrid phosphopeptide datasets (**Supplementary Fig. 3b-c**). By using duplicate runs of PC9, the hybrid library achieves the highest coverage in the number of quantified phosphopeptides and provides 39.6% and 12% more phosphopeptides than the DDA-based and DIA-based libraries, respectively, while the DIA-based library outperformed the DDA-based library (**Supplementary Fig. 3d**). Nevertheless, quantitative comparison showed high correlation ($R^2 > 0.95$) for the commonly quantified 13,292 class 1 phosphopeptides among the three libraries (**Supplementary Fig. 3e**)."

Supplementary Figure 3

Supplementary Figure 3. Performance evaluation of hybrid library and single DDA/DIA type of libraries. (a) Experimental design to generate datasets to construct hybrid, DDA or DIA library from fractionated peptides of PC9 cells, followed by single-shot DIA analysis. (b) Phosphopeptide composition of single (DDA or DIA) and hybrid library. (c) Venn diagram showing overlapping phosphopeptide in libraries. (d) Identification results from duplicate DIA runs using a library from PC9 cells. (e) Quantification correlation of 13,292 class one phosphopeptide using the different libraries.

2. Connected to point 1, lines 613-614 do not indicate FDR re-processing after merging of DDA and DIA into the hybrid library. If so, could this inflate the library with false positive IDs?

Response: Thank you for the suggestions and we fully agree that controlling FDR is crucial. To address your comment, we reconstructed the spectral library using a strategy reported previously (Muntel et al., *Mol. Omics*, 2019) and re-analyzed all the dataset in this study. We have reprocessed the DDA and DIA files (n=180 raw files) using Spectronaut Pulsar search filtering

with the same criteria at 1% FDR of PSM, peptide and protein for the GPS library. This new library consists of 159,524 phosphopeptides (121,407 class 1) corresponding to 88,107 phosphosites on 8,805 protein groups. (**Figure 3b**). For confident phosphosite determination for each precursor, the phosphosite localization tool recently reported by Bekker-Jensen et al. (*Nat. Commun* 2020) and integrated into Spectronaut was applied to filter class 1 phosphosite localization. The result has been updated in **Figure 3**.

3. The authors present both in Fig 4 and the abstract, that they merge the IDs of the dirDIA and libDIA analyses to get even higher ID analysis results. Similarly, as in point 1, could this inflate the number of false positive IDs, since results from different analysis pipelines were not subjected to a common FDR? In the worst case, could the gain of IDs over the individual dirDIA/libDIA approach be attributed to false positive IDs? (If so, the approach might still be useful to generate extremely robust IDs, by filtering for the overlap only - similar as in PeptideShaker by Vaudel et al. 2015.)

Response: Thank you for the suggestions. We would like to clarify the different analysis in this study.

3.1 In **Figure 4a-c**, we aimed to compare the identification coverage of conventional single-shot DDA and DIA, and further evaluated the performance and complementarity of library-based (*libDIA*) and direct DIA (*dirDIA*). Due to the superior identification coverage of the library-based DIA, thus, we show the overall identification result combining the library-based result and additional identification result from the direct DIA. We agree that common FDR should be performed if a user wants to combine two results, which may require additional algorithms to be developed. Thus, we reported identification results for the two approaches separately.

3.2 Combining quantification results from the two approaches with an unbiased FDR is still challenging. Therefore, throughout this study, all the other analysis including synthetic phosphopeptides (**Figure 2**), cell lines (**Figure 5**) and tissue (**Figure 6**) were all performed using library-based DIA with the same FDR control and site localization confidence.

4. The authors claim that they generated the deepest DIA spectral library for a single cancer tissue type (NSCLC). Since the plug-and-play library generated by Lawrence et al. (2016) covers substantially more phospho-peptides, could an analysis using this library reach the same depth and performance on the NSCLC samples?

Response: Thank you for the suggestions. As we discussed above, the assay is available for targeted PRM-based assay in the website of Phosphopedia (<https://phosphopedia.gs.washington.edu/PhosphoproteomicsAssay/>) and its format is not for DIA. Following the comments, we have downloaded the list of phosphopeptides reported in the article and compared our phosphopeptide spectral library with plug-and-play library (Supplementary **Fig. 4b**), showing that the GPS library in this study provides complementary identification. Nevertheless, we found that the current format of dataset in **plug-and-play library can not be used for Spectronaut.**

5. It is not always clear for which statistical tests and figures FDR control was applied. In general, statistical tests on proteomics data should always include multiple testing correction. Lines 684-688 read as if FDR control was only used for TKI test, but not for tumor-vs-normal test. Fig 5f shows $-\log(p\text{-value})$, which indicates no multiple testing correction was applied. Was FDR control not applied in these cases? If so, why not?

Response: Thank you for the comments. For the cell line experiment (**Figure 5**), the criteria used was adjusted FDR <0.01 , $S0=0.1$ from two-sample t-test. For the tissue experiment (**Figure 6**), the phosphosite abundance was normalized to the corresponding protein expression to obtain the alteration of phosphorylation level. The differential expression was determined by two-sample t-test ($p\text{-value} <0.05$, $S0=0.1$). The description on the statistical evaluation has been added in the revised manuscript.

6. The authors do not present a discussion **of limitations** in their discussion section. If they agree that some of the points raised here are valid, they might want to consider discussing them.

Response: Thank you for the suggestions. There are several challenges in DIA analysis. We have added the discussion of limitations of our approach in the revised manuscript, including the tool for FDR processing on combining *dirDIA* and *libDIA* datasets, significantly underexplored phosphoproteome in a sample-specific manner, spectra library applicable to different MS platforms, will further enhance applicability of the technology. The following revisions were added to the discussion section.

Page 22, line 492-495: *“Despite the complementary identification and quantification results offered by libDIA and dirDIA approaches, nevertheless, a FDR control to ensure the identification confidence from the merging datasets of different fragmentation patterns is essential and remains to be developed.”*

Page 23, line 518-523: *“Further advancement in DIA strategy, such as more efficient DIA acquisition modes and MS platforms, and extending spectral libraries applicable to multiple cancer types will further enhance applicability of the technology. Besides, informatics tools for error rate estimation to integrate targeted and direct DIA analysis results will further improve coverage and quantification accuracy to implement the strategy towards clinical application.”*

== Minor Comments ==

7. The authors use pervanadate to increase **phospho-tyrosine IDs**. Could it be that due to chemical induction, the gain in IDs mostly reflects phospho-peptides not relevant in biological settings? Would an enrichment using anti-phospho-tyrosine antibodies be more appropriate?

Response: We thank you for the suggestions. The enrichment with anti-phosphotyrosine antibodies is a useful strategy to enhance the identification coverage of phosphotyrosine. Though the use of pervanadate is a chemical method, it achieved the aim to obtain good quality of

phosphotyrosine mass spectra for library construction. We believe that these phosphotyrosine spectra help to enhance the identification of low abundant phosphotyrosine site. For example, without the use of anti-EGFR immunoprecipitation, we were able to identify 12 tyrosine sites of EGFR, including the autophosphorylation sites of Y1197 and Y1172 that are the two major responsive sites to activating driver mutation on EGFR. The roles of these identified phosphotyrosine sites may require further functional analysis.

8. Since the data was already analysed using Spectronaut, would it be possible to achieve the same localization performance presented by DIA phosphoRS using the Spectronaut localization approach (Bekker-Jensen et al. 2020)?

Response: We thank you for the suggestions. Accordingly, we compared the new localization approach (Bekker-Jensen et al. *Nat. Commun.* 2020) using the new Spectronaut version (V14). Using the same version for data analysis, we further compared the two approaches. The results show comparable localized phosphopeptides, of which 75% of class 1 localized phosphopeptides are in common. In the revised manuscript, we implemented the Spectronaut for site localization. Please see the following figure

Figure. Comparison of site localization by PhosphoRS and Spectronaut using PC9 and CL68 cell. (a) Class 1 localized phosphopeptide precursors. (b) Class 1 localized phosphosites from the two cell lines. Triplicate DIA files of each cell line were analyzed using Spectronaut (v14) for signal extraction against GPS library (originally reported in the manuscript). Then site localization was performed using Spectronaut and by PhosphoRS. The result indicated 72% and 75% overlap in precursor and phosphosites, respectively.

9. For the MaxQuant phospho-peptide DDA analysis, match-between-runs (MBR) was used. How was localization filtering performed here, since MBR does not provide localization information (Fig 4a)?

Response: Thank you for the suggestions. We are aware that site localization was not performed in the MBR. Here we aim to show the comparison of the high missing values in DDA even if MBR was applied. Thus, site localization was not discussed in **Figure 4**.

10. It seems that some samples were used both for library construction AND as samples themselves (e.g. tumor samples 21 and 24, supp table 1b). Were these measured as independent replicates, or could they otherwise lead to overfitting of data?

Response: Thank you for the suggestions. For hybrid library construction, the tissue samples were pooled (including tumor sample 21 and 24) and processed by peptide fractionation-based analysis to obtain DDA data sets. For DIA analysis, each tumor was analyzed independently without pooling. The detailed source and clinical information of the cell lines were shown in **Supplementary Table 1**.

11. It could be beneficial to point out that Fig 5 phosphorylation intensities are normalized on protein levels in the figure legends, since this directly influences how to interpret the data. Importantly, it is not clear in Fig 6 if this data was also protein-level normalized or not.

Response: Thank you for the suggestions. We further clarified that normalization was performed.

12. The authors might want to consider exchanging their pie chart plots (Fig 3c, 3d) with other appropriate charts (e.g. bar), as pie chart data is difficult to interpret/compare.

Response: Thank you for the suggestions. We changed the pie charts to bar graphs in **Figure 3b-c**. For **Figure 3d**, we want to show the coverage at protein level compared to the size of the pathway as well as the number of phosphosites. Thus, we used the pie chart but further revised it for better clarity.

13. Based on references in the text (lines 260 and 272), it seems that supplemental figures 4 and 5 are displayed in the wrong order.

Response: Thank you for pointing out the mistake. We have corrected the order.

14. Data import into Excel seems to have caused date conversion of gene names (e.g. Supplemental table 5a line 16599: gene “SEPT6” was renamed “6-Sept”) -> to facilitate unproblematic reanalysis of data, consider reimporting/correcting these cases (see Ziemann, Eren and El-Osta 2016)

Response: Thank you for pointing out the issue and the reference. We have corrected the text format (Supporting Table files).

15. The authors provide an extensive list of samples used, and a detailed materials and methods part. However, it would be great to also have a table listing which raw files were used for which analysis (libraries and samples) / which figure, as the raw files have arbitrary names. Additionally, it would be great to have the **custom R scripts** and **Perseus session** files published on public repositories to allow others to understand the statistical tests applied.

Response: Thank you for the suggestion. We provided a list of raw files with more details (**Supplementary Table 8**). The Perseus tool sessions were also uploaded.

16. If R packages were used that require citation, these should be cited in the material and methods section.

Response: Thank you for the suggestion. The R script for site annotation and quantification was written in-house.

REVIEWER COMMENTS

Reviewer #1 (Remarks to the Author):

The authors have addressed most of my concerns and the manuscript has been greatly improved. Nevertheless, I suggested the authors should further clarify their responses to my major concern #9 in the manuscript rather than only in the rebuttal letter. As also pointed out by the authors, a main objective in this study is to demonstrate the analytical merits and potential of DIA-based approach for large-scale analysis of clinical samples. Since TMT-based DDA approach is nowadays predominately used for clinical sample analysis, this is a very important technical question with broad interest for the whole community. In addition, the authors' conclusion that this approach "strengths in high sensitivity, deep coverage, high efficiency, low missing values as well as resolving the ratio compression problem" seems promising but somewhat overinterpreted based on the current data presented in the response letter. I also suggested the authors discuss the limitation of this approach.

Reviewer #2 (Remarks to the Author):

The authors Kitata et al present a substantially improved manuscript that has sufficiently resolved my previous concerns. The authors mention that "Custom R code used for site specific annotation and quantification is available upon request." I recommend that this also be uploaded to the JPost repository along with the published dataset.

Reviewer #3 (Remarks to the Author):

Conclusion

I would like to thank the authors for their substantial review, including both new data and new analyses. Most of my concerns have been satisfactorily addressed, except one point regarding the superiority of hybrid libraries compared to DDA / DIA only libraries (see below). I do think this point is important, since it is a major novelty of the manuscript. If it gets addressed though, I do think this manuscript should be accepted.

Major Comments

1. In my major comment 1, I was skeptical whether 1) a hybrid library does actually lead to better identification coverage / quantification, and 2) whether it could introduce its own problems, such as batch effects between identified peptide populations. I thank the authors for their extensive answer, including a new supplementary figure 3. I do agree that this figure indicates that a hybrid library is equivalent with regards to quantifications, which also should have revealed potential batch effects. However, I do not agree with the author's conclusion with regards to identifications. The authors show that a hybrid library of 14 DDA + 14 DIA runs (total 28 runs) yields more phospho-peptides than the respective DDA and DIA libraries alone (total 14 runs each). I would argue that due to the semi-stochastic nature of mass spectrometry, this is to be expected – more runs usually lead to more identifications. The real question, however, is whether a hybrid library consisting of 7 DDA + 7 DIA runs (a total of 14!) will yield more identifications than libraries consisting of 14 DDA or 14 DIA runs, respectively. This should be easy to check with the available data, but is crucial for the manuscript's assessment, whether indeed hybrid libraries are better than just DDA or DIA libraries, or whether the perceived gain in identifications is simply due to more runs used.

Minor Comments

2. I appreciate that the authors have expanded the discussion, but am a bit surprised by the statement that "FDR control to ensure the identification confidence from the merging datasets of different fragmentation patterns is essential and remains to be developed". Do the authors have evidence that merging DDA and DIA data into the hybrid library is not efficiently controlled by FDR? If so, it would be worth elaborating on this. It might also be that I am misunderstanding this sentence, in which case I apologize.

Point-by-point responses to Reviewers' comments
Manuscript number: NCOMMS-20-31521A

A Data-independent Acquisition-based Global Phosphoproteomics System Enables Deep Profiling

We thank all reviewers for their kind and constructive comments and suggestions to improve our manuscript. We have attempted to address the comments and questions either in the body of the revised manuscript, or in the point-by-point reply letter. We hope that our revisions and explanations adequately respond to the comments, clarify and enhance the manuscript. The point-by-point responses to Reviewers' comments are listed in a separate "point-by-point response" file while the revised text is highlighted in blue font.

REVIEWER COMMENTS

Reviewer #1 (Remarks to the Author):

The authors have addressed most of my concerns and the manuscript has been greatly improved. Nevertheless, I suggested the authors should further clarify their responses to my major concern #9 in the manuscript rather than only in the rebuttal letter. As also pointed out by the authors, a main objective in this study is to demonstrate the analytical merits and potential of DIA-based approach for large-scale analysis of clinical samples. Since TMT-based DDA approach is nowadays predominately used for clinical sample analysis, this is a very important technical question with broad interest for the whole community. In addition, the authors' conclusion that this approach "strengths in high sensitivity, deep coverage, high efficiency, low missing values as well as resolving the ratio compression problem" seems promising but somewhat overinterpreted based on the current data presented in the response letter. I also suggested the authors discuss the limitation of this approach.

Response: Thank you for the comment. We have added the following text and supplementary figure to the **Results** of the revised manuscript:

Page 21, line 482-499: *"To evaluate the technical advancement on the phosphoproteomics profiling for large-scale analysis of clinical samples by the GPS, the above tissue phosphoproteomics profiling results were compared to our previous tandem mass tag (TMT) phosphoproteomic datasets from 102 pairs of tumor and adjacent normal tissues from NSCLC patients (Cell 2020, 182, 226-244.e217). From combined data sets of tissue samples, <5% of 166,792 phosphopeptides were reproducibly quantified in at least 75% of the samples (0.2 mg*

per tissue) in 80 patients; a large amount of phosphoproteomics data were generated but limited number overlap across all samples. As a comparison, reproducible profiling of >30,000 phosphosites were achieved for the 5 pairs of tissues without peptide fractionation strategy. Although comparison of TMT with DIA was not the main scope of this manuscript, the comparison also demonstrated superior quantification of DIA to improve the known problem of ratio compression (tumor-to-normal). By comparing the phosphosites ratio for 5 pairs of tissues from DIA and the TMT datasets (**Supplementary Fig. 10**), the DIA results have generally larger ratios compared to these in TMT dataset. For example, EGFR-T693 known to have decreased expression in lung cancer has more obvious down-regulation in the DIA result. In summary, DIA-based label-free approach may offer an efficient alternative for large scale phosphoproteomic profiling of tissue samples requiring much lower starting amount.”

Supplementary Figure 10

Gene	Position	1260T/N		1244T/N		1326T/N		1332T/N		1259T/N	
		TMT	DIA	TMT	DIA	TMT	DIA	TMT	DIA	TMT	DIA
BRAF	S446	0.82	1.26	0.90	1.30	0.71	1.48	1.25	2.53	2.20	1.45
EGFR	T693	0.89	0.51	0.68	0.15	0.32	0.22	0.95	0.41	0.56	0.42
PML	S403	1.03	2.13	1.23	0.41	1.34	1.64	0.70	0.75	1.22	1.65
CTND1	S252	0.98	0.50	0.90	0.85	0.78	0.77	0.82	0.49	1.60	1.72
ERBB3	S686	1.49	1.28	0.54	0.22	0.56	0.74	1.10	0.80	0.72	0.39
ERBB3	S982	1.15	0.85	0.41	0.11	0.54	0.47	0.84	0.39	0.80	0.31
PML	S518	0.92	0.66	0.93	1.22	1.04	1.24	0.85	0.98	1.88	2.29

Supplementary Figure 10. Comparison of phosphosite ratios (tumor/normal) obtained by tandem mass tag (TMT) and DIA approaches. The TMT data was obtained from Chen et.al. (Cell 2020, 182, 226-244.e217). Red and green color shows upregulation and down-regulation, respectively.

We thank the Reviewer to remind us of the conclusion of our method. The statement of “strengths in high sensitivity, deep coverage, high efficiency, low missing values as well as resolving the ratio compression problem” was shown only in the previous point-by-point letter and not in the Manuscript. For your reference, the conclusion of our method is described as follows:

Page 17, line 380-383: “These results highlighted the advantages of DIA to allow deep profiling and highly reproducible quantification between runs, which are critical benefits for multiplexed quantification, such as clinical proteomics, for many specimens.”

Page 22, line 497-499: “In summary, DIA-based label-free approach may offer an efficient alternative for large scale phosphoproteomic profiling of tissue samples requiring much lower starting amount.”

We also discussed the limitation in the **Discussion**:

Page 24, line 543-548: “Further advancement in DIA strategy, such as more efficient DIA acquisition modes and MS platforms, and extending spectral libraries applicable to multiple

cancer types will further enhance applicability of the technology. Besides, informatics tools for error rate estimation to integrate targeted and direct DIA analysis results will further improve coverage and quantification accuracy to implement the strategy towards clinical application."

Reviewer #2 (Remarks to the Author):

The authors Kitata et al present a substantially improved manuscript that has sufficiently resolved my previous concerns. The authors mention that "Custom R code used for site specific annotation and quantification is available upon request." I recommend that this also be uploaded to the jPOST repository along with the published dataset.

Response: Thank you for the comment. We uploaded the R code for site quantification to the jPOST database with the file name "Rcode-site-quantification."

The description is added in the Code availability section of the revised manuscript.

Page 38, line 858-859: *"Custom R code used for site quantification based on site annotation provided by Spectronaut was deposited in the jPOST repository."*

Reviewer #3 (Remarks to the Author):

Conclusion

I would like to thank the authors for their substantial review, including both new data and new analyses. Most of my concerns have been satisfactorily addressed, except one point regarding the superiority of hybrid libraries compared to DDA / DIA only libraries (see below). I do think this point is important, since it is a major novelty of the manuscript. If it gets addressed though, I do think this manuscript should be accepted.

Major Comments

1. In my major comment 1, I was skeptical whether 1) a hybrid library does actually lead to better identification coverage / quantification, and 2) whether it could introduce its own problems, such as batch effects between identified peptide populations. I thank the authors for their extensive

answer, including a new supplementary figure 3. I do agree that this figure indicates that a hybrid library is equivalent with regards to quantifications, which also should have revealed potential batch effects. However, I do not agree with the author's conclusion with regards to identifications. The authors show that a hybrid library of 14 DDA + 14 DIA runs (total 28 runs) yields more phospho-peptides than the respective DDA and DIA libraries alone (total 14 runs each). I would argue that due to the semi-stochastic nature of mass spectrometry, this is to be expected – more runs usually lead to more identifications. The real question, however, is whether a hybrid library consisting of 7 DDA + 7 DIA runs (a total of 14!) will yield more identifications than libraries consisting of 14 DDA or 14 DIA runs, respectively. This should be easy to check with the available data, but is crucial for the manuscript's assessment, whether indeed hybrid libraries are better than just DDA or DIA libraries, or whether the perceived gain in identifications is simply due to more runs used.

Response: Thank you for the comment to further clarify the strength of hybrid library, if any. Following your suggestion, we further evaluated in detail the performance of hybrid library at both proteome (HeLa lysate digest) and phosphoproteome (PC9 & CL68 cell, Fe-IMAC enriched) datasets. In both cases, following the suggestions from review comments, equal numbers of fractionated peptide LC-MS/MS runs were used for DDA, DIA-based library, 16 runs for HeLa and 14 runs for PC9 and CL68 (**Supplementary Fig. 7a**). For hybrid library, one of the duplicate runs of each fraction were selected to provide equal number of raw files resulting in equal number as single type libraries. Then a separate triplicate runs of single-shot DIA of each cell lines were used for quantification performance evaluation. The new result indicated that hybrid library size increased, likely due to the complementary identification from DDA and DIA. For single-shot DIA analysis, the hybrid library also resulted in the highest number of peptides and phosphopeptides in all three cell lines with high quantification correlation (**Supplementary Fig. 7b-d**).

Accordingly, the revised **Supplementary Fig. 7** is provided including the following revised description in the text.

Page 14, line 306-325: *“The performance of a hybrid library was evaluated by comparing the spectra library constructed from a single acquisition type of DDA or DIA. Two datasets of comparable size were generated from DDA and DIA using three cell lysate digest coupled with StageTip fractionation: proteome from HeLa and phosphoproteome from PC9 and CL68 cell lines (**Supplementary Fig. 7a**). Three sets of proteome and phosphoproteome spectra libraries were independently constructed by using DDA, DIA and hybrid datasets of equal numbers of raw files. For the library construction from HeLa lysate (n=16), the number of peptides in the hybrid library increased by 8% and 22% compared to DDA and DIA, respectively, showing the expected complementary nature of merging DDA and DIA datasets in the hybrid library (**Supplementary Fig. 7b**). By using another triplicate runs of single shot DIA dataset from HeLa cells, the hybrid*

library achieved the highest coverage with 12% and 5% more quantified peptides compared to the DDA-based and DIA-based libraries, respectively, while the DIA-based library outperformed the DDA-based library (Supplementary Fig. 7b). Similarly, for phosphoproteome dataset from PC9 and CL68, the hybrid libraries (n=14) also resulted in the highest number of phosphopeptides (Supplementary Fig.7c). Besides, the single shot DIA data mapping to the DIA-based library still outperformed the result from the DDA-based library likely due to the similar nature of fragmentation pattern. Nevertheless, quantitative comparison showed high correlation ($R^2 > 0.9$) for the quantified phosphopeptides among the three libraries (Supplementary Fig. 7d)."

Supplementary Figure 7

Supplementary Figure 7. Comparison on the performance of hybrid library, DDA-based and DIA-based libraries. (a) Experimental design to generate hybrid library from fractionated peptides library construction followed by single-shot DIA analysis. (b) Proteome library constructed from HeLa cells and DIA quantitation against DDA, DIA and hybrid libraries. (c) Phosphopeptides library and DIA quantitation against DDA, DIA and hybrid libraries for PC9 and CL68 cells. (d) Quantification correlation of phosphopeptides from the different libraries using $\text{Log}_2(\text{peak area})$ of phosphopeptides. Spectral libraries were constructed from 16 raw files of fractionated HeLa peptides for each DDA, DIA and hybrid libraries. Phosphopeptides libraries were constructed from 14 raw files of PC9 and CL68 cells for each DDA, DIA and hybrid libraries. Triplicate single-shot DIA was then acquired to evaluate quantification performance of the libraries.

Minor Comments

2. I appreciate that the authors have expanded the discussion, but am a bit surprised by the statement that “FDR control to ensure the identification confidence from the merging datasets of different fragmentation patterns is essential and remains to be developed”. Do the authors have evidence that merging DDA and DIA data into the hybrid library is not efficiently controlled by FDR? If so, it would be worth elaborating on this. It might also be that I am misunderstanding this sentence, in which case I apologize.

Response: Thank you for the comment. In this study two steps of data processing were involved. In the first step, hybrid library was generated from DDA and DIA datasets. The second step will process the single shot DIA dataset by direct DIA and library-based DIA. By using the function in Spectronaut in our previous revision, stringent FDR control can be achieved for merging the DDA and DIA datasets in library generation. However, the processed datasets cannot be merged due to the lack of FDR control from the different and complementary data processing tools that can easily combine the search outputs of DIA datasets. So, the statement in the Discussion indicates the lack of FDR control in the second step, where merging the direct DIA and library-based DIA results for enhanced identification still requires development of tools for FDR estimation.

We have revised the manuscript accordingly to clarify the statement:

Page 23, line 517-520: *“Despite the complementary identification and quantification results offered by libDIA and dirDIA approaches, nevertheless, a FDR control to ensure the identification confidence from the merging DIA output of different analysis approaches or software is essential and remains to be developed.”*